# Genomic signatures of pre-resistance in *Mycobacterium tuberculosis*

Arturo Torres Ortiz [1], Jorge Coronel[2], Julia Rios Vidal[3], Cesar Bonilla[3,4], David A. J. Moore [5], Robert H. Gilman[6], Francois Balloux [7], Onn Min Kon[8], Xavier Didelot [9] & Louis Grandjean [1,10 ✉]

Recent advances in bacterial whole-genome sequencing have resulted in a comprehensive catalog of antibiotic resistance genomic signatures in *Mycobacterium tuberculosis*. With a view to pre-empt the emergence of resistance, we hypothesized that pre-existing polymorphisms in susceptible genotypes (pre-resistance mutations) could increase the risk of becoming resistant in the future. We sequenced whole genomes from 3135 isolates sampled over a 17-year period. After reconstructing ancestral genomes on time-calibrated phylogenetic trees, we developed and applied a genome-wide survival analysis to determine the hazard of resistance acquisition. We demonstrate that *M. tuberculosis* lineage 2 has a higher risk of acquiring resistance than lineage 4, and estimate a higher hazard of rifampicin resistance evolution following isoniazid mono-resistance. Furthermore, we describe loci and genomic polymorphisms associated with a higher risk of resistance acquisition. Identifying markers of future antibiotic resistance could enable targeted therapy to prevent resistance emergence in *M. tuberculosis* and other pathogens.

[1] Imperial College London, Department of Infectious Diseases, London, UK. [2] Universidad Peruana Cayetano Heredia, Lima, Perú. [3] Unidad Técnica de Tuberculosis MDR, Ministerio de Salud, Lima, Perú. [4] Universidad Privada San Juan Bautista, Lima, Perú. [5] London School of Hygiene and Tropical Medicine, London, UK. [6] Johns Hopkins Bloomberg School of Public Health, Baltimore, MD, USA. [7] UCL Genetics Institute, London, UK. [8] Respiratory Medicine, National Heart and Lung Institute, Imperial College London, London, UK. [9] University of Warwick, School of Life Sciences and Department of Statistics, Warwick, UK. [10] UCL Department of Infection, Institute of Child Health, London, UK. ✉email: l.grandjean@ucl.ac.uk

**M**ycobacterium tuberculosis is estimated to have killed 1 billion people over the last 200 years[1] and remains one of the world's most deadly pathogens[2]. Drug resistance in bacteria, particularly the *Enterobacteriaceae* and *Mycobacterium tuberculosis*, imposes an unsustainable burden on health programs worldwide with some strains so extensively resistant that they are untreatable with existing antibiotic therapy[3]. Although recent advances in bacterial whole-genome sequencing have significantly improved the identification of drug resistance[4], post hoc approaches to diagnosis miss the opportunity to preempt the emergence of drug resistance and implement preventive measures prior to the acquisition and spread of antibiotic resistant disease.

An increased risk of drug resistance emergence is often attributed to inadequate implementation of control measures[5], but bacterial factors have also been proposed as potential contributors to drug resistance[6]. Evidence of differential drug resistance acquisition at the *M. tuberculosis* sublineage level is conflicting. Epidemiological and in vitro studies have suggested that the Beijing family, belonging to lineage 2, is hyper-mutable[7] with a propensity to develop resistance at a higher frequency than other lineages[8–11], while others cite evidence to the contrary[12,13]. Pre-existing resistance to one antibiotic (mono-resistance) is another factor that may influence the acquisition of multidrug-resistance[14]. Mono-resistance to isoniazid or rifampicin has been associated with increased rates of multidrug-resistance acquisition[15,16], but the relative risk of either remains unclear. Similarly, phylogenetic analyses suggest a stepwise progression towards multidrug-resistance, where mutations conferring isoniazid resistance tend to precede those linked to rifampicin resistance[17–20].

Phylogenetic trees have been increasingly used to study pathogen dynamics and evolutionary processes of a wide range of phenotypes of epidemiological interest, including virulence and drug resistance acquisition[21,22]. A necessary focus on improving the molecular diagnosis of drug resistance has led to the generation of large strain collections of drug resistant pathogens. However, unrepresentative samples of this kind enriched for drug-resistant isolates limit the ability to characterize the evolution and dynamics of drug resistance from a diverse background of ancestral susceptible strains. Inadequate sampling without comprehensive population level coverage or sufficient temporal span compounds this problem, while the monomorphic nature of the *M. tuberculosis* genome makes constructing time-calibrated phylogenetic trees particularly challenging. As a consequence, a single mutation rate is often applied to the data, but this assumption inappropriately forces lineages and sub-lineages to conform to the same global mutation rate, thus limiting the inferences that can be made from the data.

Overcoming these issues, we present findings from samples collected over a 17-year time span with population level coverage in the hyperendemic suburbs of Lima, Peru. We apply a genome-wide survival analysis to a time-calibrated phylogeny of 3135 *M. tuberculosis* strains, and show the existence of pre-resistance mutations among drug susceptible genotypes that increase the risk of future drug resistance emergence in *M. tuberculosis*. We demonstrate significant differences in the acquisition of drug resistance between lineages, on mono-resistant backgrounds, and at the level of nucleotide polymorphisms. Our findings were then tested and replicated in an independent publicly available data set of 1027 whole genomes collected in Samara, Russia, and in a collection of 1573 isolates from multiple countries to demonstrate that they can be globally generalized.

## Results

**Population structure, genomic analysis, and patient demographics.** A total of 3432 *M. tuberculosis* genomes from Lima (Peru) were analyzed, of which 3135 passed genomic quality filters. Of this, 2037 were part of a population level study carried out in 2009 where sputum samples were taken from all patients presenting tuberculosis symptoms in the Lima areas of Callao and Lima South[23] (Supplementary Data File 1). Comparison of drug resistance prevalence between the population level sampling using molecular genotyping and reports of epidemiological data in Peru[2] are consistent: 1.5% (32/2037) of samples were rifampicin mono-resistant; 5% were isoniazid mono-resistant (105/2037), and 13% were multidrug-resistant (251/2037) (Supplementary Table 1). The remaining samples were collected from cohort studies covering a 17-year period of research in the regions of Lima and Callao in order to achieve a sufficient temporal span in our sampling window (Supplementary Fig. 1). Both lineage 2 and lineage 4 had a similar distribution of sampling dates (Supplementary Fig. 2).

The isolates were first aligned to the reference genome H37Rv, then lineages and sublineages were assigned using clade specific SNPs[24]. Lineage 4 (L4, Euro-American) consisted of 2807 samples, while lineage 2 (L2, Beijing) had 327 isolates (Table 1). There was a single representative of lineage 1 (Indo-Oceanic), which was used to root the phylogenetic tree. The remaining samples from the data set, which included 5 *M. caprae* isolates, were not used in the downstream analysis. Lineage 4 had the highest diversity, comprising 1235 isolates from lineage 4.3 (LAM), 935 from lineage 4.1.2.1 (Haarlem), 271 from lineage 4.1.1 (X-Type), and 312 from lineages denoted as Type T, which encompasses lineages 4.5, 4.7, 4.8 and 4.9. Other minor sublineages included lineage 4.2.2 (TUR) and lineage 4.4. All isolates from Lineage 2 were part of the Beijing sublineage (lineage 2.2), mainly from the sublineage 2.2.1 or Modern Beijing, and with only one representative of the sublineage 2.2.2 or Asia Ancestral[25] (Table 1).

The alignment of the isolates to the reference genome resulted in 64,586 SNPs, of which 18,022 were singletons (28%). Most SNPs were not widely distributed across the population, and only 8088 variants had a frequency in the dataset higher than 1%. A total of 59,789 SNPs (16,934 singletons, 28%) were identified for lineage 4, and 4821 SNPs (1370 singletons, 28%) for lineage 2. We applied the same analysis to a publicly available data set of 1027 isolates from Samara, Russia, as a validation set where, unlike the

---

**Table 1 Population structure.**

|  | Clade name | Number |
|---|---|---|
| **Lineage 1** | Indo-Oceanic | 1 |
| **Lineage 2** | East-Asian | 327 |
| Lineage 2.2.1 | Beijing | 319 |
| Lineage 2.2.1.1 | Beijing | 7 |
| Lineage 2.2.2 | Beijing | 1 |
| **Lineage 4** | Euro-American | 2807 |
| Lineage 4.1.1 | X Type | 271 |
| Lineage 4.1.2.1 | Haarlem | 935 |
| Lineage 4.2.2 | TUR | 3 |
| Lineage 4.3.1 | LAM | 5 |
| Lineage 4.3.2 | LAM3 | 328 |
| Lineage 4.3.3 | LAM9 | 676 |
| Lineage 4.3.4 | LAM11 | 226 |
| Lineage 4.4 | - | 51 |
| Lineage 4.5 | T Type | 4 |
| Lineage 4.7 | T Type | 31 |
| Lineage 4.8 | T Type | 86 |
| Lineage 4.9 | T Type | 12 |
| Lineage 4[1] | T Type | 179 |

Lineages and sublineages defined using clade specific SNPs[24].
[1]Clade name assigned phylogenetically.

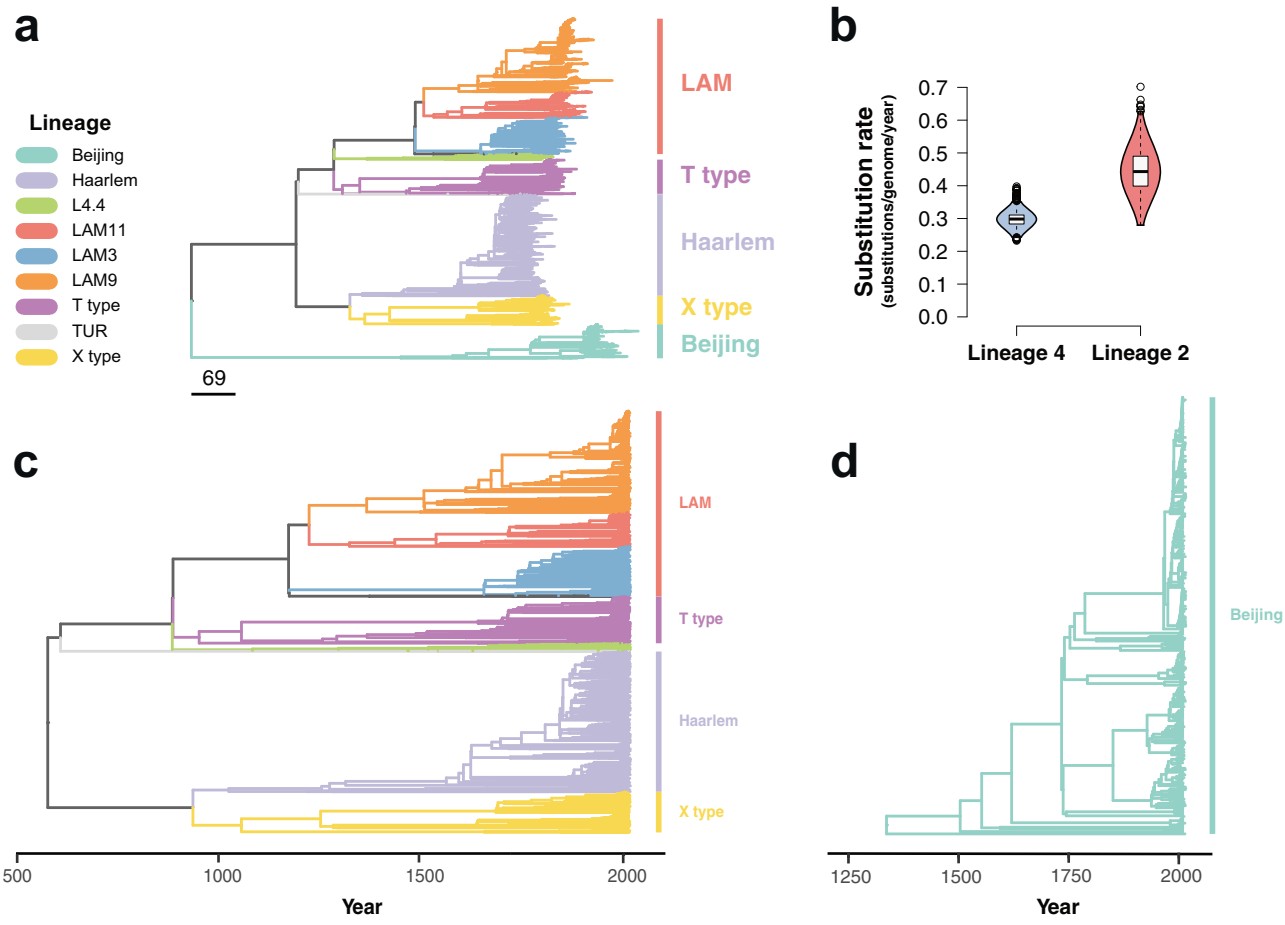

**Fig. 1 Phylogenetic analysis of 3134 *Mycobacterium tuberculosis* isolates from Lima, Peru.** Colors represent different lineages and sublineages. **a** Maximum likelihood phylogeny. Scale in number of substitutions per genome. **b** Violin plots showing the posterior density distribution of the inferred substitution rate in substitutions per genome per year derived by sampling from 10[7] MCMC iterations. The substitution rate was estimated separately for lineage 4 (blue) and lineage 2 (red). Box plots inside the violin indicate the median value of the distribution (black horizontal line) and the interquartile range. Whiskers denote 1.5x the interquartile range, while outliers are plotted as individual points. **c** Time-calibrated phylogeny of lineage 4. **d** Time-calibrated phylogeny of lineage 2.

Peruvian dataset, lineage 2 constitutes the main lineage. We identified a total of 28,414 SNPs, consistent with previous publications of this data[19].

Patient demographic metadata was available for 2220 samples, of which 88% were smear positive, 27% were previously treated for tuberculosis, and 2.8% were HIV positive, consistent with previous population level estimates in Peru[26]. The median age was 28 years (IQR 21–41). In our cohort, 86% of lineage 2 and 89% of lineage 4 were smear positive.

**Phylogenetic analysis and drug resistance emergence.** The maximum likelihood phylogeny constructed using these alignments grouped the isolates by lineage similarly to previously published global data sets[24] (Fig. 1a, Supplementary Fig. 3). To study the temporal dynamics of drug resistance acquisition at the population level, the maximum likelihood phylogeny was time-calibrated using the sampling dates of the isolates, which extended from 1999 to 2016. Dated phylogenies were built separately for lineage 2 and lineage 4 in order to avoid the confounding effect of the temporal and population structures[27]. Before time calibration of the phylogeny, we tested the adequacy of the temporal correlation of evolutionary change to reliably infer the model parameters. First, a root-to-tip linear regression of the number of substitutions from the root and the sampling times was fitted to confirm a positive association between time and

evolutionary change. As uneven sampling may bias the root-to-tip regression[28], a date-randomization test was additionally performed using the full Bayesian model implemented in BactDating[29] with the original dataset and 100 randomizations where the sampling times were permuted, representing the expectations of the model parameters in the absence of temporal signal. The substitution rate estimated for the original dataset and for the 100 randomizations was compared to verify a lack of overlap between the 95% credible intervals. Both lineage 4 and lineage 2 datasets showed a clear temporal signal (Supplementary Fig. 4), and thus model parameters could be confidently inferred from the data[30,31]. We used the relaxed clock model implemented in BactDating[29], allowing the mutation rate to vary in each branch independently. We ran the MCMC until convergence of the chains was achieved, with an effective sample size (ESS) of at least 100 (Supplementary Fig. 5). The estimated rate for lineage 2 was 0.45 substitutions per genome per year (0.32–0.57 95% CI), while lineage 4 had a clock rate of 0.299 (0.25–0.36 95% CI) (Fig. 1b). The estimates of the molecular clock for both lineages were consistent with previous reports[32]. The most common recent ancestor (tMRCA) for our samples was placed at 565 CE (263;826 95% CI) for lineage 4 (Fig. 1c), while lineage 2 had a tMRCA in 1325 CE (1070;1499 95% CI) (Fig. 1d).

Drug resistance was inferred for all isolates at the tips of the phylogenetic tree using well-established drug resistance associated

**Table 2 First emergence of drug resistance conferring mutations in Lima, Peru.**

| Drug | Lineage | Gene | Year | Mutation |
|---|---|---|---|---|
| RIF | Lineage 4 | rpoB | 1951.7 [1931.7–1970.7] | p.Ser450Leu |
| | Lineage 2 | rpoB | 1974.4 [1953.2–1986.8] | p.Ser450Leu |
| INH | Lineage 4 | KatG | 1941.6 [1913.6–1959.7] | p.Ser315Thr |
| | Lineage 2 | KatG | 1957.5 [1928.3–1977.4] | p.Ser315Thr |
| ETH | Lineage 4 | embB | 1973.7 [1967.8–1980.8] | p.Gly406Ala |
| | Lineage 2 | embB | 1983.1 [1967.0–1994.1] | p.Met306Val |
| PZA | Lineage 4 | pncA | 1962.4 [1942.6–1972.2] | p.His51Arg |
| | Lineage 2 | pncA | 2002.1 [1997.0–2005.5] | c.-11A>C |
| STR | Lineage 4 | rpsL | 1974.5 [1955.9–1986.2] | p.Lys43Arg |
| | Lineage 2 | rpsL | 1958.1 [1934.6–1975.1] | p.Lys43Arg |

Emergence of drug resistance conferring mutations for the 5 antibiotics historically used as first line drugs for the treatment of tuberculosis. *RIF* rifampicin, *INH* isoniazid, *ETH* ethambutol, *PZA* pyrazinamide, *STR* streptomycin. Year presented as a point estimate with the highest posterior density interval.

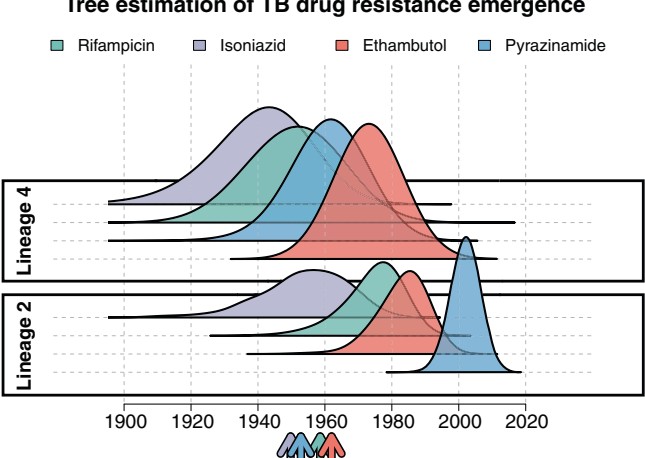

**Tree estimation of TB drug resistance emergence**

Rifampicin ■ Isoniazid ■ Ethambutol ■ Pyrazinamide

**Introduction of new TB compounds**

**Fig. 2 Inferred posterior density distribution of the earliest occurrence of resistance to first line antituberculous drugs.** Posterior density distribution inferred using a time-calibrated phylogeny for both lineage 4 and lineage 2. Arrows represent the approximate time of antibiotic introduction.

SNPs[33]. In addition to molecular typing of drug resistance, all isolates included in the analysis had Drug Susceptibility Testing (DST) performed either by MODS or by the proportional method in agar. DST and molecular typing showed consistent results for 96% of the samples for rifampicin resistance and 92% for isoniazid resistance. The time of emergence of drug resistant mutations was inferred by reconstructing the ancestral sequences of the internal nodes in the phylogenetic tree. The time of emergence of a specific antibiotic resistance mutation was approximated to the inferred year of the internal node where such mutation first appeared. The phylogenetic estimates of the emergence of drug resistance conferring mutations in lineage 4 occurred around the time of the known introductions of the corresponding drug. In contrast the emergence of drug resistance in lineage 2 was observed to have arisen many years after the introduction of antituberculous drugs. This is consistent with the geographic spread of lineage 4 in Europe together with early widespread use of drugs in this region (Table 2, Fig. 2). For both lineage 2 and lineage 4, the earliest inferred occurrence of resistance was to isoniazid, by the Ser315Thr mutation in the gene KatG, around 1957 (1928;1978 95% CI) for lineage 2, and 1942 (1913;1960 95% CI) for lineage 4, in line with the reported wide introduction of isoniazid in 1952[34]. The rifamycins were

first isolated in 1957[34], and we estimate the date for the first acquisition of resistance to rifampicin due to the rpoB mutation Ser450Leu to have emerged around 1951 (1931;1971 95% CI) for lineage 4 and 1974 (1953;1988 95% CI) for lineage 2. None of the drug resistant nodes reverted to susceptible along the branches of the two phylogenetic trees.

**The emergence of compensatory mutations.** It has been shown that secondary mutations arising after the acquisition of drug resistance may alleviate the fitness cost associated to antibiotic resistance mutations[35], but little is known about their temporal dynamics. Data on *M. tuberculosis* drug resistance compensatory mechanisms is mainly limited to isoniazid and rifampicin resistance[36,37]. Non-synonymous mutations in the gene *rpoC* have been suggested as secondary compensatory mutations for rifampicin associated mutations in the *rpoB* gene[37]. A total of 34% (258/755) of lineage 4 isolates harboring *rpoB* mutations also had *rpoC* non-synonymous polymorphisms; for lineage 2, 38% (33/87) of isolates with *rpoB* mutations carried *rpoC* polymorphisms. No significant differences were observed between lineages in a logistic regression model (OR = 0.85, 95% CI 0.54–1.35, *p*-value = 0.49). Overall, 62% of rifampicin resistant isolates carried Ser450Leu rpoB mutations (525/842). Rifampicin resistant isolates harboring Ser450Leu rpoB mutations had a higher probability of carrying mutations in the *rpoC* gene (52%, 272/525) than isolates with other rpoB mutations (6%, 19/317) in a logistic regression model (OR = 16.86, 95% CI 10.55–28.50, *p*-value = $4 \times 10^{-29}$). Only 3% (9/291) of the isolates carried two non-synonymous mutations in the *rpoC* gene, while the rest had only one.

To understand the emergence of *rpoC* non-synonymous mutations, we scanned the phylogenetic branches of rifampicin resistant isolates from the root to the tip, using the inferred sequences of the ancestral nodes to determine the time of emergence of *rpoC* non-synonymous mutations. The analysis was repeated in 100 bootstrap phylogenies to infer confidence values around our estimates. In both lineage 2 and lineage 4, the emergence of *rpoC* non-synonymous mutations occurred immediately after or at the same time as the emergence of rifampicin resistance, and continued steadily over time (Fig. 3). For lineage 2, there was not a single emergence of *rpoC* mutations occurring prior to the rifampicin resistance conferring mutations. In the case of lineage 4, two *rpoC* mutations emerged before rifampicin resistance: c.765150 G > A and c.765590 C > A. Both mutations appeared once independently in the entire phylogeny. The mutation c.765150 G > A emerged in our dataset around the year 936 CE (727;1110 95% CI), in the MRCA of the clades X-type and Haarlem, all of which present the c.765150 G > A mutation (1206/2807 isolates). For c.765590 C > A, the estimated

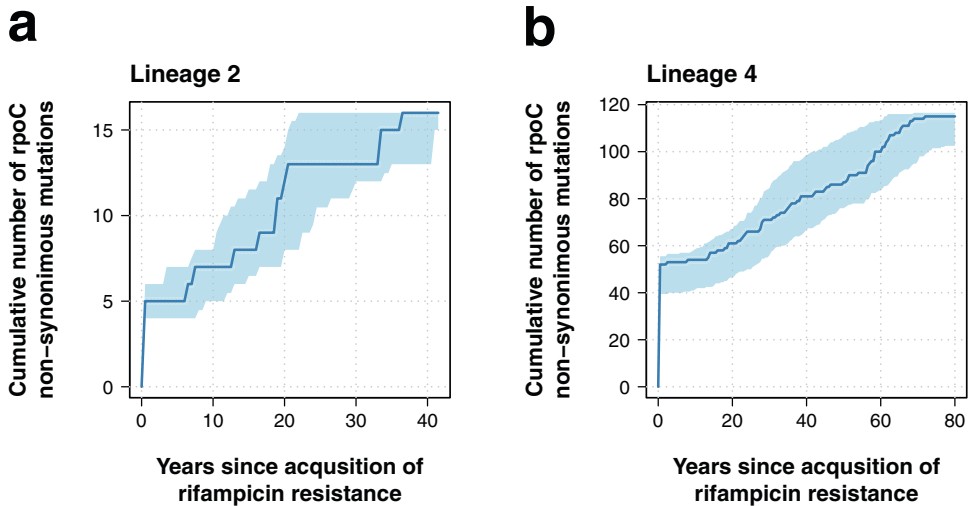

**Fig. 3 Dynamics of non-synonymous mutations in *rpoC*. a**, **b** Cumulative number of non-synonymous mutations in *rpoC* over time. The *x*-axis represents the years since the inferred time of rifampcin resistance (time 0). Dark blue line shows the cumulative number of mutations for the ML tree, while the 95% confidence interval (shaded area) is inferred by repeating the analysis in 100 bootstrap phylogenies. The analysis was performed separately for **a** lineage 2 and **b** lineage 4.

year of emergence was around 1867 CE (1813;1907 95% CI), and all the tips harboring the mutation belonged to a cluster of isolates part of the LAM11 sublineage (33/2807 isolates). Given that both mutations showed a clear phylogenetic structure and emerged independently only once in our dataset, they were considered phylogenetic mutations and were removed from the final analysis. The emergence of non-synonymous mutations in the *rpoC* gene was similar for isolates carrying the Ser450Leu rpoB mutation and for those isolates with other rpoB mutations (Supplementary Fig. 6).

**Phylogeographic history of *Mycobacterium tuberculosis* in Peru**. To estimate the year of *M. tuberculosis* introductions to Peru, we subsampled the Peruvian isolates and analyzed them alongside global representatives of both lineage 2 and lineage 4 for which both collection date and geographic origin were known (Supplementary Data File 2). The phylogenies were time-calibrated using a relaxed clock model as implemented in BactDating[29], and MCMC convergence was assessed using the traces of the model parameters (Supplementary Fig. 7).

The phylogeographic history was inferred by reconstructing the ancestral states by maximum likelihood. The geographical origin of the isolates was treated as a discrete character, and we assumed that the time of introduction occurred at the first Peruvian node of each clade.

Two early introductions of lineage 2 from China in 1880 (1856;1905 95% CI) and 1906 (1881;1929 95% CI) accounted for 84% of the Peruvian dataset (275/327 isolates), and 77% (90/117) of the drug resistance isolates as well as 77% (20/26) of the independent resistance evolutionary events (Fig. 4a). Later introductions to Peru from China occurred between 1981 (1969;1989 95% CI) and 2004 (1999;2007 95% CI), with one introduction in 1986 (1974;1994 95% CI) from South Asia representing 6 isolates, one of which is drug resistant.

Lineage 4 was inferred to have been introduced in Peru several times over the years, mainly from Europe and Brazil (Fig. 4b). Lineage 4.3 (LAM) represents the first and main introduction from Brazil around 1512 (1383;1598 95% CI), shortly after the well documented arrival of the Europeans to the continent. Subsequent smaller introductions from Europe and Brazil around 1644 (1545;1713 95% CI) and 1743 (1671;1796 95% CI) shaped

the LAM and T type lineages. We inferred that most of L4.1.2.1, part of the Haarlem sublineage, likely evolved from two introductions from Europe around 1693 (1614;1749 95% CI) and 1834 (1781;1869 95% CI). The most likely introductions of the X type clade (L4.1.1) occurred from Brazil in 1692 (1597;1754 95% CI) and South Africa for L4.1.1.3 in 1706 (1615;1773 95% CI).

**Between lineage differences in drug resistance acquisition**. The risk of acquiring drug resistance was calculated as the Cox Proportional Hazard Ratio (HR) using the time between sensitive internal nodes and the first drug resistant node in the time-calibrated phylogenetic trees. The Kaplan–Meier curves showed that lineage 2 had a higher probability of acquiring drug resistance than lineage 4 (Log-rank test *p*-value = $1.2 \times 10^{-9}$; Fig. 5a). The estimated hazard ratio of drug resistance acquisition for lineage 2 was estimated to be 3.36 when compared to lineage 4 (HR 3.36, 95% CI 2.10–5.38, Likelihood ratio test *p*-value = $4.25 \times 10^{-7}$). A similar trend was observed in the Samara dataset (HR 4.82, 95% CI 3.74–6.21, Likelihood ratio test *p*-value = $6.8 \times 10^{-34}$; Kaplan–Meier curve Log-rank test *p*-value = $1 \times 10^{-39}$; Fig. 5b). The risk of drug resistance acquisition was also higher in lineage 2 when compared to all sublineages of lineage 4 in the Peruvian dataset, using LAM3 as a reference (lineage 2 HR 3.32, 95% CI 1.84–6.28, Likelihood ratio test *p*-value = $1.9 \times 10^{-4}$, all other *p*-values > 0.2; Kaplan–Meier curve Log-rank test *p*-value = $6.9 \times 10^{-8}$; Fig. 5c). To assess the adequacy of our data to the proportional hazard assumption, we calculated the relationship between the Schoenfeld residuals against time. In all cases, a non-significant association between the Schoenfeld residuals and time supported the use of the proportional hazards model (Supplementary Fig. 8).

In order to evaluate the robustness of our maximum likelihood phylogeny, we repeated both the dating and the survival analysis on 100 phylogenetic bootstrap replicates. In both lineage 2 and lineage 4, the Cox Proportional Hazard ratio was not significantly different between the maximum likelihood phylogeny and the bootstrap replicates (Supplementary Fig. 9a, b). Additionally, the Kaplan-Meier curve of the 100 replicates was similar to that of the maximum likelihood tree (Supplementary Fig. 9c, d).

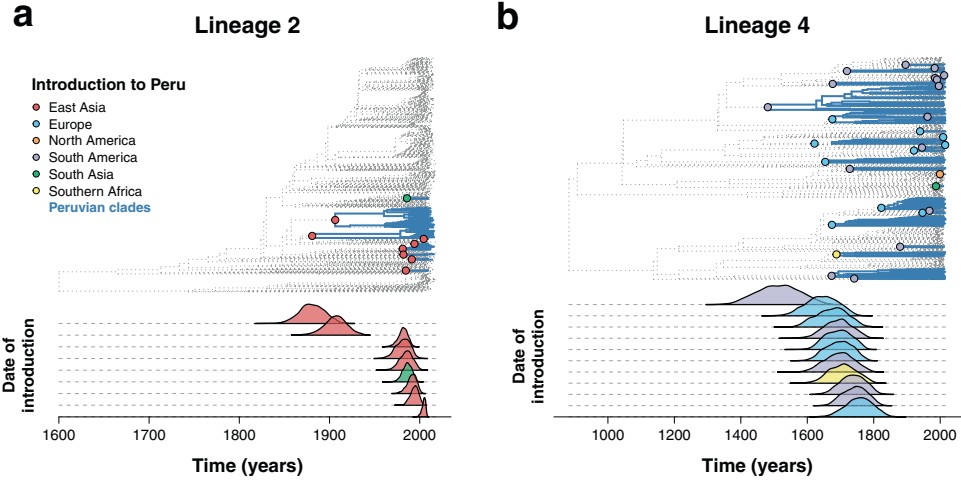

**Fig. 4 Phylogeographic analysis of *Mycobacterium tuberculosis* introductions to Peru. a, b** Inferred introductions of *Mycobacterium tuberculosis* in Peru. The top part shows a time-calibrated phylogeny, with inferred introductions to Peru highlighted in the nodes with colors representing the country from which the clade was introduced. Peruvian clades are shown in blue. The bottom part shows the estimated year of introduction. The analysis was done separately for **a** lineage 2 and **b** lineage 4. For visual representation purposes, only the year of introduction of clades with more than 10 tips are shown for lineage 4.

Both phylogenetic trees were subsampled to include only the isolates from the 2009 population level study to prevent any biases due to inclusion of datasets enriched for drug resistance isolates (Supplementary Table 1). The results were congruent with those obtained with the entire dataset. Lineage 2 was characterized by a higher risk of drug resistance acquisition when compared to lineage 4 (HR 4.84, 95% CI 2.78–8.45, Likelihood ratio test $p$-value $= 2.7 \times 10^{-8}$; Kaplan-Meier curve Log-rank test $p$-value $= 7.9 \times 10^{-10}$; Supplementary Fig. 10a). Moreover, lineage 2 also had a higher hazard ratio than any sublineage of lineage 4 (lineage 2 HR 5.1, 95% CI 2.17–11.9, Likelihood ratio test $p$-value $= 1.8 \times 10^{-4}$, all other $p$-values $> 0.2$; Kaplan-Meier curve Log-rank test $p$-value $= 3.02 \times 10^{-7}$; Supplementary Fig. 10b).

Several confounders may also explain the differential rate of drug resistance acquisition between lineages (Supplementary Fig. 11). We found no significant association between *M. tuberculosis* sublineages and HIV (Supplementary Fig. 11a) and smear positivity (Supplementary Fig. 11b) in a logistic regression model ($n = 2133$, all $p$-values $> 0.1$). It has been previously shown that prison conditions increase and amplify drug-resistance tuberculosis in Peru[38,39]. Our study population showed a higher proportion of prison infection in lineage 2 (5.8%, 12/206 patients) when compared to lineage 4 (1.4%, 28/2011 patients, Supplementary Fig. 11c), and a higher risk of lineage 2 infection within the prison population ($n = 2217$, OR $= 4.4$, 95% CI 2.1–8.5, $p < 0.001$). This finding did not explain the higher rate of drug resistance acquisition in lineage 2, since all the lineage 2 samples taken from prisoners belonged to the same cluster and none of them harbored drug resistance conferring mutations. Since previous treatment with antituberculous drugs has been associated to an increased risk of acquiring drug resistance[38], we tested the association between previous treatment and the different sublineages of our cohort. Previous treatment history was available for 2236 samples contained metadata regarding previous treatment of tuberculosis. We found no significant association between *M. tuberculosis* sublineages and previous treatment with antituberculous drugs in a logistic regression model ($n = 2236$, all $p$-values $> 0.1$), suggesting that the between lineage differences in drug resistance acquisition observed in the survival analysis are not confounded by a differential distribution

of antibiotics between sublineages (Supplementary Fig. 11d). Behavioral patterns may also affect the dynamics of drug resistance acquisition and transmission. Patient sex (Supplementary Fig. 11e) was not significantly associated with any of the *M. tuberculosis* sublineages ($n = 2205$, $p$-value $= 0.3$). The mean age for patients with lineage 4 infection was 33.3 (22–41 IQR), while the mean age for patients with lineage 2 was 30.4 (21–36 IQR). Although the distribution of patient age was similar between the two lineages (Supplementary Fig. 11f), lineage 4 showed an incident rate of age 1.09 higher than lineage 2 in a quasi-Poisson model (lineage 4 estimate $= 0.09$, standard error $= 0.02$, $p = 0.001$). On the other hand, age was not significantly associated with a higher risk of drug resistance acquisition in a logistic model (OR 0.999, 95%CI 0.994–1.003, $p = 0.69$).

**The risk of developing MDR TB from isoniazid mono-resistance**. To determine the effect of mono-resistance on the acquisition of further multidrug-resistance, the hazard ratio of acquiring rifampicin resistance was calculated for isoniazid mono-resistant ancestral genotypes versus susceptible ancestral strains. Genotypes with an isoniazid mono-resistant background had 15 times the hazard of developing rifampicin resistance tuberculosis relative to wild type susceptible strains (HR 15.12, 95% CI 10.54–21.69, Likelihood ratio test $p$-value $< 10^{-15}$; Kaplan–Meier curve Log-rank test $p$-value $= 2, 7 \times 10^{-63}$; Fig. 6a). A larger hazard ratio was obtained in the Samara dataset (HR 37.28, 95% CI 18.81–73.88, Likelihood ratio test $p$-value $= 3.4 \times 10^{-25}$; Kaplan–Meier curve $p$-value $= 4.6 \times 10^{-63}$; Fig. 6b), although the low prevalence of mono-resistance clades in the Samara set may bias this estimate.

Multidrug-resistance was preceded by rifampicin mono-resistance only one time in the phylogenetic tree. Due to the infrequent occurrence of rifampicin mono-resistance prior to multidrug-resistance emergence, the risk of developing multidrug-resistance from a rifampicin mono-resistance background could not be reliably estimated.

**Genomic signatures of drug resistance acquisition**. Genome-wide survival analysis was performed using a Cox Proportional Hazard regression model to identify genetic variants in phylogenetic

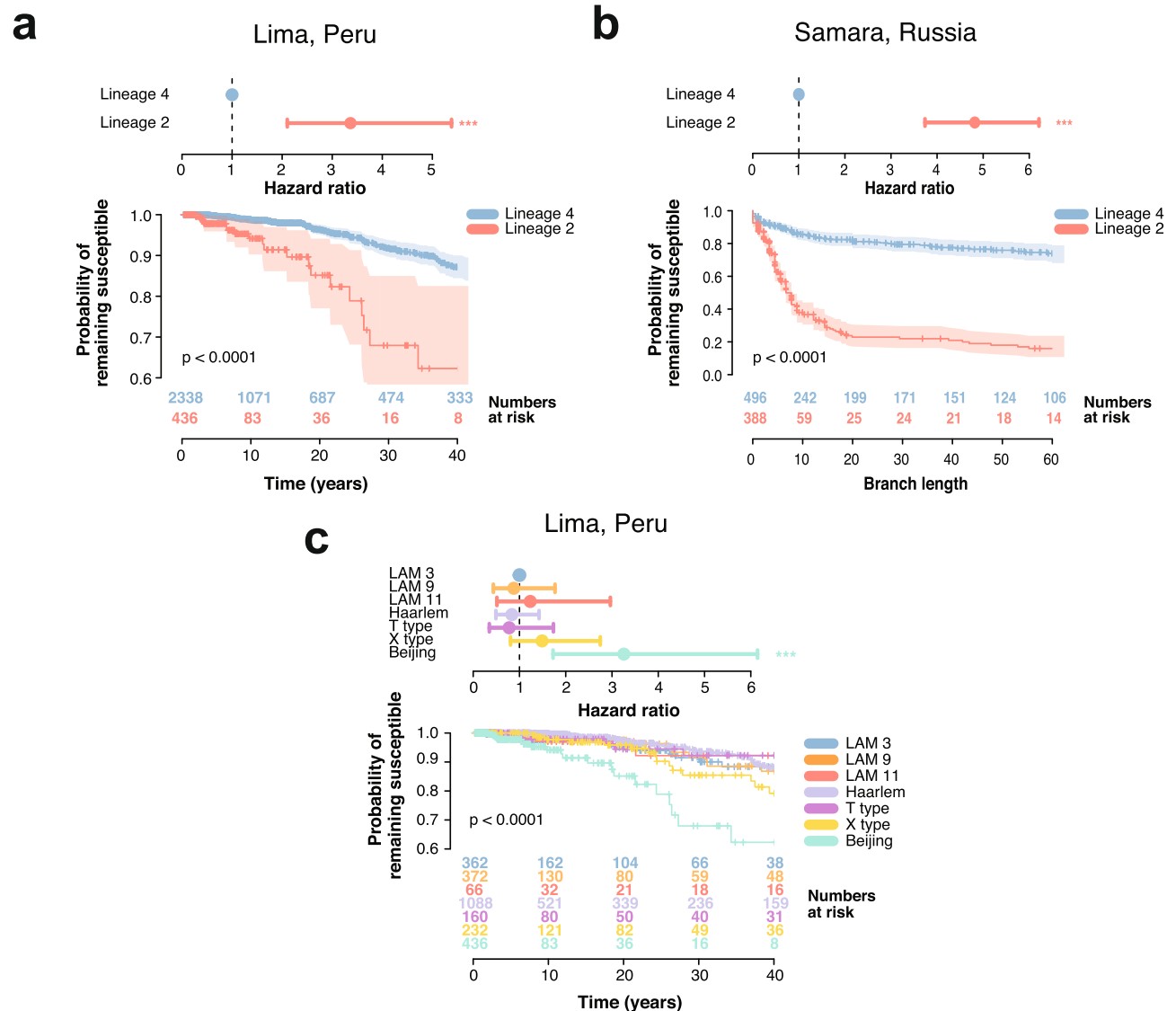

**Fig. 5 Hazard ratio and Kaplan–Meier curve for different sublineages of *Mycobacterium tuberculosis*. a–c** Top: Hazard ratio (HR). Points and error bars represent the HR estimate and the 95% CI, respectively. The *p*-value for the HR was calculated using the likelihood ratio test. Bottom: Kaplan–Meier curve and numbers at risk. Y-axis represents the probability of remaining susceptible to any antibiotic, while the X-axis shows the time in years or the distance in branch length. Shaded areas show the 95% CI. Kaplan–Meier curves were compared and *p*-values were derived using the log-rank test. **a** Depicts the HR of lineage 2 compared to lineage 4 in the Peruvian dataset (HR 3.36, 95% CI 2.10–5.38, Likelihood ratio test *p*-value = $4.25 \times 10^{-7}$) and the different Kaplan–Meier curve for lineage 2 and lineage 4 (Log-rank test *p*-value = $1.2 \times 10^{-9}$). **b** Same metrics for the Samara dataset (HR 4.82, 95% CI 3.74–6.21, Likelihood ratio test *p*-value = $6.8 \times 10^{-34}$; Kaplan–Meier curve Log-rank test *p*-value = $1 \times 10^{-39}$). **c** Shows HR between lineage 2 and the different sublineages of lineage 4 found in the Peruvian dataset (LAM9, LAM3, LAM11, Haarlem, X type and T type), using LAM3 as a reference (lineage 2 HR 3.32, 95% CI 1.84–6.28, Likelihood ratio test *p*-value = $1.9 \times 10^{-4}$, all other *p*-values > 0.2; Kaplan–Meier curve Log-rank test *p*-value = $6.9 \times 10^{-8}$). Statistical significance of the hazard ratio differences presented next to the CI bars (\**p* < 0.05; \*\**p* < 0.01; \*\*\**p* < 0.001).

nodes inferred to be drug susceptible but associated with a higher risk of progression towards drug resistance. Resistant nodes were defined as those inferred to be resistant to any antibiotic in order to identify common pathways of increased risk of acquiring resistance regardless of the specific antibiotic. The association analysis was performed in lineage 4 and lineage 2 separately, and we further corrected for population structure using a kinship matrix, which reduced the genomic inflation factor ($\lambda$) to 1.16 (Supplementary Fig. 12).

Six variants in drug susceptible ancestral genotypes were associated with a higher risk of acquiring drug resistance in lineage 4 after population and multiple testing correction, three of which were in previously annotated genes (Fig. 7a). The variant with the lowest p-value corresponded to a 9 bp deletion at

location 2,604,157 in the locus *lppP*, which encodes a lipoprotein and has been predicted to be required for growth in macrophages[40]. This deletion had a frequency of 1.7% in the population, and it evolved 12 times independently along the phylogenetic tree. Genotypes with this variant had a hazard ratio 7.36 times greater than those with an intact *lppP* (Fig. 7b, HR 7.36 95% CI 3.85–14.04, *p*-value = $7.46 \times 10^{-10}$). We replicated our findings in a global data set of 1573 L4 isolates (Supplementary Data File 3), which was relatively enriched for drug resistance (55% of samples were resistant to any drug). The *lppP* deletion had a frequency of 9% and inferred susceptible genotypes with the deletion had a hazard ratio 3.6 times greater than those without it (HR 3.6, 95% CI 1.9–6.9, *p*-value = $8.7 \times 10^{-5}$). Two synonymous polymorphisms at *esx* genes were found to be

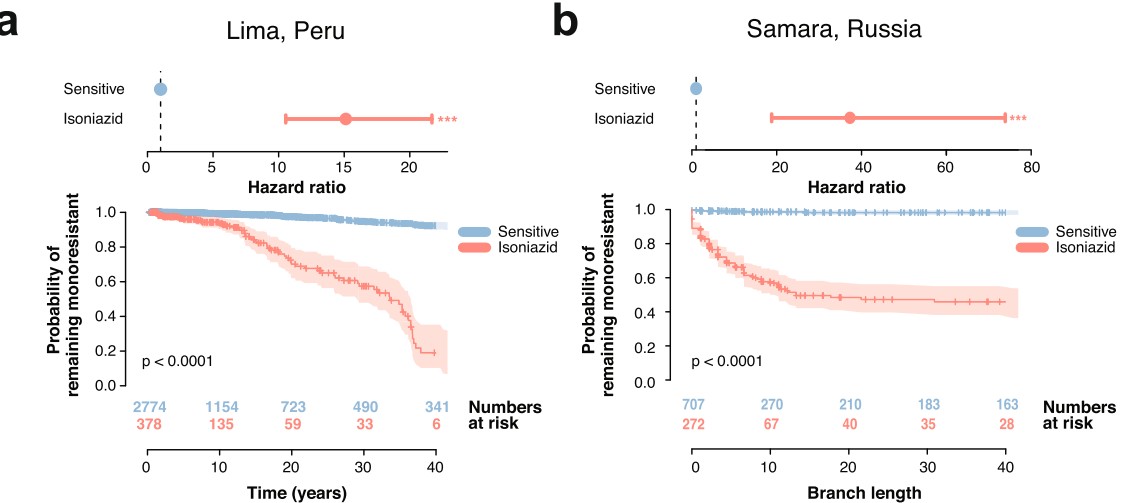

**Fig. 6 Hazard ratio and Kaplan–Meier curve for rifampicin acquisition. a, b** Top: Hazard ratio (HR). Points and error bars represent the HR estimate and the 95% CI, respectively. The *p*-value for the HR was calculated using the likelihood ratio test. Bottom: Kaplan–Meier curve and numbers at risk. Y-axis represents the probability of remaining susceptible to rifampicin, while the X-axis shows the time in years or the distance in branch length. Shaded areas show the 95% confidence interval. *P*-values for the Kaplan–Meier curves were calculated using the log-rank test. **a** Depicts the risk of acquiring rifampicin resistance from an already isoniazid mono-resistant background compared to a drug susceptible one (HR 15.12, 95% CI 10.54–21.69, Likelihood ratio test *p*-value = $1.3 \times 10^{-40}$) and the Kaplan–Meier curves for the different backgrounds (Log-rank test *p*-value = $2.7 \times 10^{-63}$). **b** Same metrics for the Samara dataset (HR 37.28, 95% CI 18.81–73.88, Likelihood ratio test *p*-value = $3.4 \times 10^{-25}$; Kaplan–Meier curve *p*-value = $4.6 \times 10^{-63}$). Statistical significance of the hazard ratio differences presented next to the CI bars (*$p < 0.05$; **$p < 0.01$; ***$p < 0.001$).

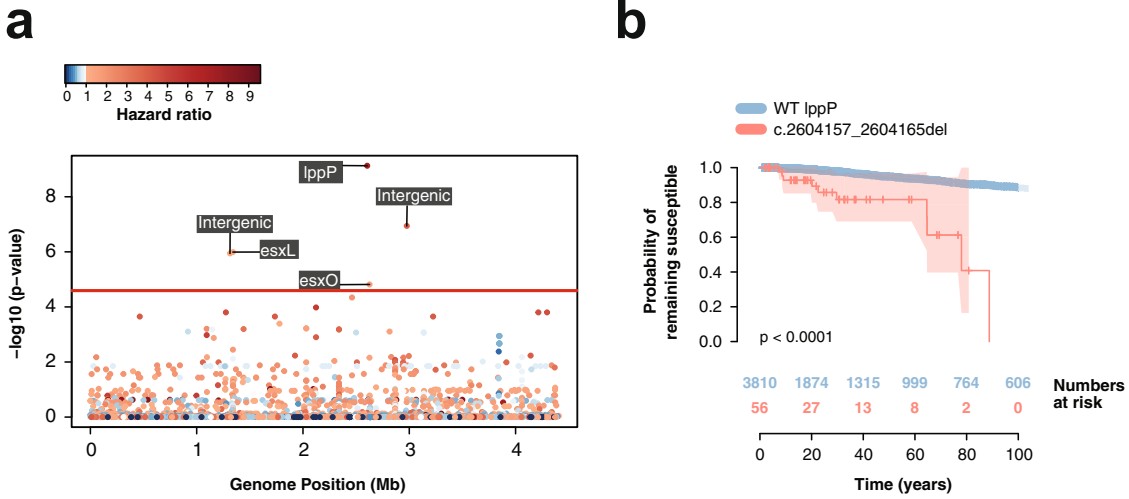

**Fig. 7 Genome-Wide association study (GWAS) results. a** Manhattan plot for GWAS on increased risk of drug resistance acquisition in lineage 4. The red line represents the Bonferroni corrected *p*-value threshold of $3.37 \times 10^{-5}$. Labels show the genes where the significant hits are located. Colors indicate the hazard ratio, with a scale of blue representing hazards ratio lower than 1 and a scale of reds for hazard ratios higher than 1. **b** Kaplan–Meier curve and numbers at risk of a 9 bp deletion in the gene *lppP* comparing the probability of remaining susceptible between those nodes without the deletion (blue) and those with it (red). Shaded areas represent the 95% CI. The *p*-value for the Kaplan–Meier curves was calculated using the log-rank test.

associated with a higher risk of acquiring drug resistance in inferred drug susceptible genotypes. The *esx* gene family encodes protein secretion systems described to be critical for growth, pathogenesis, and mycobacterial–host interactions[41]. The two polymorphisms were detected in the gene *esxL* at position 1,341,044 with a hazard ratio of 3.2 (HR 3.2 95% CI 1.91–5.37, *p*-value = $1.01 \times 10^{-6}$), and at position 2,626,011 in the gene *esxO* (HR 11.12, 95% CI 5.50–22.5 *p*-value = $1.52 \times 10^{-5}$) with a frequency in the population of 17 and 5%, respectively. In the L4 global dataset, the *esxL* SNP had a frequency of 19.5% while the *esxO polymorphism* was present in 10% of the isolates. Inferred susceptible genotypes in the global data set carrying the mutation in *esxO* had a risk of acquiring drug resistance 3.1 times higher

than those with the reference genotype (HR 3.1, 95% CI 1.3–7.3, *p*-value = 0.009), while those carrying the mutation in *esxL* had a risk 1.4 higher, although differences where not statistically significant (HR 1.4, 95% CI 0.7–3.0, *p*-value = 0.3). Visual inspection of the short-read alignments around the described genes was undertaken to confirm high quality alignments over these regions (Supplementary Fig. 13).

For the gene-based GWAS, non-synonymous variants were aggregated for each locus and a binary matrix was created reflecting whether internal nodes and tips contained at least one non-synonymous polymorphism for each gene. After population and Bonferroni multiple testing correction, a total of 35 variants had a *p*-value lower than the significance threshold of $4.38 \times 10^{-5}$

**Table 3 Gene-based association analysis.**

| Gene Name | Rv Number | HR | SE | P-value[1] | Frequency | Functional category[2] |
|---|---|---|---|---|---|---|
| Rv2510c | Rv2510c | 16.74 | 0.4 | $1.79 \times 10^{-14}$ | 0.01 | Unknown |
| yrbE2A | Rv0587 | 4.77 | 0.39 | $2.43 \times 10^{-12}$ | 0.01 | Virulence, detoxification, adaptation |
| kdpA | Rv1029 | 1.46 | 0.81 | $3.56 \times 10^{-10}$ | 0.42 | Cell wall and cell processes |
| lppP | Rv2330c | 7.14 | 0.45 | $4.44 \times 10^{-10}$ | 0.02 | Cell wall and cell processes |
| cfp2 | Rv2376c | 1.97 | 1.04 | $8.4 \times 10^{-09}$ | 0.03 | Cell wall and cell processes |
| lytB1 | Rv3382c | 5.2 | 0.45 | $3.05 \times 10^{-08}$ | 0.01 | Cell wall and cell processes |
| frdB | Rv1553 | 1.62 | 0.64 | $3.54 \times 10^{-08}$ | 0.42 | Metabolism and respiration |
| mmpL1 | Rv0402c | 1.6 | 0.38 | $1.07 \times 10^{-07}$ | 0.06 | Cell wall and cell processes |
| fadD5 | Rv0166 | 2.27 | 0.47 | $1.57 \times 10^{-07}$ | 0.05 | Lipid metabolism |
| cyp135A1 | Rv0327c | 2.07 | 0.4 | $2.03 \times 10^{-07}$ | 0.04 | Metabolism and respiration |
| Rv1897c | Rv1897c | 1.18 | 0.72 | $2.55 \times 10^{-07}$ | 0.16 | Unknown |
| Rv3113 | Rv3113 | 1.69 | 0.72 | $2.55 \times 10^{-07}$ | 0.52 | Metabolism and respiration |
| gpdA2 | Rv2982c | 0.9 | 0.69 | $2.99 \times 10^{-07}$ | 0.05 | Lipid metabolism |
| Rv0579 | Rv0579 | 1.33 | 0.53 | $6.7 \times 10^{-07}$ | 0.02 | Unknown |
| Rv1417 | Rv1417 | 0.83 | 0.72 | $9.51 \times 10^{-07}$ | 0.16 | Cell wall and cell processes |
| recD | Rv0629c | 2.63 | 0.37 | $1 \times 10^{-06}$ | 0.03 | Information pathways |
| Rv3903c | Rv3903c | 1.14 | 0.28 | $1.05 \times 10^{-06}$ | 0.19 | Unknown |
| hemE | Rv2678c | 1.28 | 0.55 | $2.18 \times 10^{-06}$ | 0.02 | Metabolism and respiration |
| lgt | Rv1614 | 0.83 | 0.64 | $2.25 \times 10^{-06}$ | 0.17 | Cell wall and cell processes |
| pyrG | Rv1699 | 4.1 | 0.32 | $3.15 \times 10^{-06}$ | 0.04 | Metabolism and respiration |
| galK | Rv0620 | 0.87 | 0.72 | $4.73 \times 10^{-06}$ | 0.16 | Metabolism and respiration |
| Rv2915c | Rv2915c | 4.8 | 0.36 | $7.24 \times 10^{-06}$ | 0.02 | Unknown |
| Rv0226c | Rv0226c | 2.87 | 0.41 | $7.73 \times 10^{-06}$ | 0.03 | Cell wall and cell processes |
| Rv0021c | Rv0021c | 1.04 | 0.62 | $8.6 \times 10^{-06}$ | 0.06 | Unknown |
| caeA | Rv2224c | 0.92 | 1.05 | $8.69 \times 10^{-06}$ | 0.02 | Cell wall and cell processes |
| Rv1501 | Rv1501 | 2.26 | 0.53 | $1.09 \times 10^{-05}$ | 0.08 | Unknown |
| icd1 | Rv3339c | 4.42 | 0.53 | $1.14 \times 10^{-05}$ | 0.04 | Metabolism and respiration |
| Rv1096 | Rv1096 | 3.05 | 0.64 | $1.38 \times 10^{-05}$ | 0.02 | Metabolism and respiration |
| fum | Rv1098c | 0.9 | 0.48 | $1.56 \times 10^{-05}$ | 0.05 | Metabolism and respiration |
| galE2 | Rv0501 | 2.84 | 0.51 | $1.66 \times 10^{-05}$ | 0.01 | Metabolism and respiration |
| Rv1751 | Rv1751 | 4.43 | 0.42 | $1.67 \times 10^{-05}$ | 0.02 | Metabolism and respiration |
| murF | Rv2157c | 2.38 | 0.54 | $2.78 \times 10^{-05}$ | 0.03 | Cell wall and cell processes |
| Rv3712 | Rv3712 | 0.93 | 0.58 | $3 \times 10^{-05}$ | 0.05 | Metabolism and respiration |
| asnB | Rv2201 | 3.62 | 0.48 | $3.14 \times 10^{-05}$ | 0.02 | Metabolism and respiration |
| pstB | Rv0933 | 1.49 | 0.52 | $4.19 \times 10^{-05}$ | 0.07 | Cell wall and cell processes |

[1]P-values derived from the Likelihood ratio test. [2]Annotations extracted from the Mycobrowser[42].

(Table 3). Functional annotations curated in the Mycobrowser[42] showed that most genes were related to metabolism and cellular respiration (*frdB, cyp135A1, Rv3113, hemE, pyrG, galK, icd1, Rv1096, fum, galE2, Rv1751, Rv3712, asnB*) as well as cell wall processes (*kdpA, lppP, cfp2, lytB1, mmpL1, Rv1417, lgt, Rv0226c, caeA, murF, pstB*).

No significant associations were identified for lineage 2 after correcting for population structure, possibly due to the lower diversity of lineage 2 and the strong lineage effect on the phenotype. The analysis could not be replicated in the Samara dataset as the Samara dataset was significantly smaller and hence lacked sufficient statistical power.

## Discussion

This study represents the largest population level genomic analysis of *Mycobacterium tuberculosis* to date. Our 17-year sampling time frame provided a unique opportunity to study drug resistance acquisition dynamics and evolution. To our knowledge this is the first description and evaluation of pathogen pre-resistance (pre-existing polymorphisms that predispose to the acquisition of future drug resistance).

Using an ancestral state genome-wide survival analysis to move in time through the phylogenetic tree, we show that *M. tuberculosis* is predisposed to acquire drug resistance mutations at the lineage level, after mono-resistance, and at the level of nucleotide polymorphisms. Identifying pathogen genetic factors that predispose strains to evolve drug resistance could help prevent the acquisition and spread of resistance as well as treatment failure by expanding treatments to those strains most likely to become resistant in the future.

Previous studies of acquired drug resistance at the sublineage level in *M. tuberculosis* have led to contradictory outcomes, with small sample sizes in fluctuation assays[7,12] or using amalgamated sub-population level samples. Here we demonstrate that lineage 2 acquired resistance to antibiotics more rapidly than lineage 4. There were no significant differences observed in drug resistance acquisition between the sub-lineages of the most diverse lineage 4. Even though lineage 2 showed an increased risk in drug-resistance acquisition, lineage 4 evolved resistance earlier than lineage 2 for almost all drugs analyzed, with the exception of streptomycin. This may be explained by the Euro-American distribution of lineage 4 and the earlier widespread implementation of antibiotics in these regions. Our analysis also suggests that the acquisition dynamic of compensatory mutations was similar for both lineage 2 and lineage 4. After rifampicin resistance associated mutations evolve in a clade, non-synonymous mutations in *rpoC* start to occur and steadily accumulate over time. Thus, pre-resistance mutations emerge independently of compensatory mutations.

The identification and control of mono-resistant strains is a key component of tuberculosis public health infection prevention and control efforts. Mono-resistance is associated with worse

clinical outcomes[43] and an increased probability of progressing to multidrug-resistance[15]. At the population level, our study quantifies this risk and shows that isoniazid mono-resistant strains have at least 15 times the hazard of developing multidrug-resistance relative to wild type susceptible strains. Despite the use of new therapeutics, multi-drug resistant tuberculosis continues to require polypharmacy with increased toxicity and longer treatment duration[2]. Globally, molecular rapid drug resistance surveillance is focused primarily on rifampicin, with the widely implemented GeneXpert MTB/RIF PCR based assay unable to detect isoniazid mono-resistance. Although Drug Susceptibility Testing (DST) is the current gold standard for identification of drug resistance isolates and can detect isoniazid mono-resistance, known diagnostic delays associated with it may limit its use in reducing mono-resistance amplification[44,45]. Inadequate diagnosis of isoniazid mono-resistance will inevitably lead to inappropriate treatment and could fuel rapid evolution of multidrug-resistance, thus posing a significant threat to tuberculosis control.

We also identified loci associated with higher risk of future drug resistance acquisition. To remain polymorphic, these variants must be under balancing selection and only be positively selected once exposed to drug therapy. Rather than causing resistance directly, these variants could promote resistance acquisition by compensating for the fitness costs of resistance in vivo[35] or by increasing drug tolerance[46].

At the gene level, most non-synonymous mutations associated with pre-resistance genotypes were located in genes related to cell wall processes and metabolism. Functional studies and prospective clinical trials are warranted to confirm their association with future drug resistance acquisition.

The variant with the lowest p-value corresponded to a 9 bp deletion in the gene lppP present at a frequency of 1.7% in the population and arose 12 times independently. Deletions in lipoproteins have been well characterized in the past[47], and lppP has been predicted to be required for growth in macrophages[40]. Lipoproteins can act as antigenic proteins[47], and thus deletions in the genes encoding them may alter the interaction between the bacilli and the macrophages, potentially conferring a selective advantage in the presence of drug and increasing the probability of acquiring drug resistance. This variant was also present in a global dataset for L4 at a frequency of 9%, which could be explained by the higher prevalence of drug resistance isolates in most publicly available data sets.

Two additional synonymous variants were identified in the genes esxL and esxO, which encode the ESAT-6-like proteins esxL and esxO. These genes are part of a family of genes that encode immunogenic secreted proteins that play a role in mycobacterial growth, pathogenesis, and host-pathogen interactions[41]. Moreover, esxO has been associated with pathogenesis by inducing autophagy in infected macrophages[48]. Synonymous homoplastic variants in esx genes have been previously identified[23], but their phenotypic effects are still unclear.

This study benefited from an unbiased population level coverage of both drug resistant and drug susceptible strains that enabled us to reliably correct for the founder effect and control for the influence of pre-existing population diversity. The large sampling size and time frame—a consequence of 17-years of continued research in the same location—allowed us to generate time-calibrated phylogenies without imposing a global mutation rate. This was a pre-requisite for the downstream analyses and our GWAS survival analysis approach. We were also able to replicate our sublineage and mono-resistance dependent hazards of acquired resistance in the smaller Samara dataset. However, the time scale and size of this publicly available data was insufficient to allow us to confirm the effect of the lppP deletion in a second independent dataset.

Although our phylogenetic analysis reveals trends of drug resistance acquisition over evolutionary time, prospective cohort studies are required to determine the effect of these mutations at the individual patient and household level. Non-bacterial factors are unlikely to influence our findings, since they would have to have been disproportionately and consistently associated to a specific lineage over long periods of time. Nevertheless, we explored the influence of confounding variables on our dataset. There was no difference observed in the proportion of patients receiving previous antituberculous treatment between the two lineages. This makes our findings unlikely to be influenced by differential exposure to drugs among lineages. There was no difference in sputum smear grade between lineages, suggesting that our findings are not a consequence of increased pathogenicity of lineage 2 in comparison to lineage 4. Moreover, other factors that may affect the rates of drug resistance acquisition such as HIV status, sex, or imprisonment, did not show differences between lineages. Even though the age of patients with lineage 4 infection was significantly higher than that of patients with lineage 2, the difference was small. Additionally, patient age was not associated to a higher incidence of drug resistance, and therefore it is unlikely that age differences explain the higher risk of acquiring antibiotic resistance of lineage 2. Differential healthcare systems influence the acquisition and transmission of drug resistance tuberculosis, and thus importation events to Peru of resistant strains from specific lineages could have affected the dynamics of drug resistance acquisition. We showed that lineage 2 in Peru is characterized by two importations around 1900 CE, which is consistent with major immigration events of laborers from China to Peru at the end of the 19th century to work in the railroads, guano mines, and cotton and sugarcane plantations[49,50]. Conversely, the majority of lineage 4 clades were imported from Europe and Brazil between the 16th and 18th centuries, compatible with European colonial expansion[51]. Therefore, significant immigration to Peru occurred well before the advent of antibiotics, which limits the influence of imported drug resistant strains. Moreover, the majority of introductions occurring in recent times were of drug susceptible clades. Nevertheless, it is possible that some resistance events may have arisen as a result of importation of resistant strains from countries with different drug selection pressures.

In summary, this population wide 17 year-long epidemiological study of M. tuberculosis genetics provides the first description and evaluation of pre-resistant polymorphisms in susceptible genotypes that predispose to the acquisition of future drug resistance. Prediction of future drug resistance in susceptible pathogens together with targeted expanded therapy has the potential to prevent drug resistance emergence in M. tuberculosis and other pathogens. Prospective cohort studies of participants with and without these polymorphisms should be undertaken to inform clinical trials of personalized pathogen genomic therapy. This ancestral state genome-wide survival analysis could also be employed to predict and prevent the emergence of resistance or indeed any important trait of interest in other organisms.

## Methods

**Ethics approval**. Ethical approval for sample collection and processing was obtained from the Institutional Review Board of Universidad Peruana Cayetano Heredia and the Peruvian Ministry of Health for all individual studies from which this data was derived[38].

**Study design and sample selection**. Samples were selected from previous projects taking place across the region of Lima. The first project consisted of a population level study carried out between 2008 and 2010 as part of the population level implementation of Microscopic Observation Drug Susceptibility (MODS) testing[52,53]. A total of 2139 unselected patients of tuberculosis were collected (Supplementary Data File 1). Of these patients, 284 were analyzed in previous

studies (PRJEB5280)[23], while 1855 were processed as part of this project (PRJEB39837).

A second set of 213 MDR-TB samples was obtained from a 3-year long household follow-up study conducted between 2010–2013[38], of which 185 randomly selected samples underwent whole-genome sequencing (PRJEB5280)[23].

Additionally, 42 unpublished whole-genome sequences of samples collected from 1999 to 2007 in different regions of Lima were added to the study (PRJEB47846), as well as 575 samples that were collected between 2003 and 2013 as part of the CRyPTIC Consortium (PRJEB32234)[54], and 489 samples from the TANDEM Consortium (PRJEB23245)[55] taken between 2014 and 2016.

All samples without collection date, as well as reference clinical samples, were excluded from the final list. Drug Susceptibility Testing (DST) was performed either by MODS[53] or by the proportions method on agar[56].

To replicate our findings, all the analyses were repeated in a publicly available independent dataset of 1027 isolates from Samara, Russia (PRJEB2138), as the sampling was also representative of the population[19]. We also included a global data set of 1573 publicly available lineage 4 samples relatively enriched for drug resistance (Supplementary Data File 3).

**Whole-genome sequence analysis.** Quality analysis of the raw reads was performed using FastQC[57]. A de-novo assembly of the short reads was done with SPAdes genome assembler v3.14.0[58] across kmers of size 21, 33, 45, 55, 65, 75, 81, 101, 111, and 121. The resulting assembly contigs were mapped to the well annotated H37Rv reference genome (Gene bank: AL123456) using minimap2[59] with the asm20 option. Single nucleotide polymorphisms (SNPs) and small insertions and deletions (indels) were identified with BCFtools mpileup and BCFtools call v1.9[60] using the multiallelic calling algorithm, keeping the information about every single site in the genome in a VCF file. Lastly, indels were left-aligned and normalized using BCFtools norm. A consensus sequence was created from the VCF file. In order to determine the quality of the variants, the raw reads were mapped against the resulting consensus sequence using the mem algorithm implemented in BWA v0.7.17[61], after which the alignments were sorted using SAMtools v1.9[62] and filtered for possible PCR and optical duplicates using Picard v2.19.0[63]. Local realignment around indels was performed using the GATK v3.8-1-0 'IndelRealigner' module[64]. The mean coverage for each sample was calculated as the number of mapped bases (excluding soft-clipped bases) divided by the genome size. Samples with a mean coverage lower than 15x were excluded from subsequent analysis. SNPs and indels were detected as described in the previous step. Variants that did not meet the quality criteria were filtered using a combination of BCFtools filter and custom scripts in Python v3.7.3 with the following cutoffs: minimum Phred-scaled quality score (QUAL) of 20; minimum mapping quality (MQ) of 20; minimum genotype quality (QG) of 20; minimum read position bias (RPB), mapping quality bias (MQB), and strand bias (SP) of 0.001; minimum depth (DP) of 10 and a maximum of 5 times the mean coverage; minimum of reads supporting the alternate allele (AD) of 75% of the total depth at that position, with no less than two reads in the forward (ADF) and the reverse (ADR) strands. Additionally, SNPs within 2 bp of an indel and indels within 3 bp of another indel were removed, as both situations can be indicative of mapping artifacts. Positions that did not meet the quality criteria were annotated using the IUPAC ambiguity codes[65]. Samples with more than 25 high quality heterozygous calls were removed to avoid the inclusion of putative mixed infections. Variants that overlapped 100 bp intervals around known hypervariable regions, such repetitive elements and transposases[66], were removed from the analysis as this may affect the reliability of the alignment. Similarly, recombinant regions in genes coding for proline-glutamate (PE) or proline-proline-glutamate (PPE)[67], and SNPs implicated in drug resistance[33] were excluded in order to minimize homoplasies that could disrupt the tree topology. The resulting sequences were concatenated to generate a multiple sequence alignment. Sites with a proportion of ambiguous bases higher than 10% were excluded from the analysis. Last, samples with a proportion of ambiguous sites in the alignment higher than 5% were excluded.

The functional consequence of variants was assessed using the Variant Effect predictor (VEP) v104.3[68].

**Phylogenetic analyses.** All the phylogenetic analyses were performed based on the alignment containing both lineage 4 and lineage 2 samples, as well as separately for each lineage.

A maximum likelihood phylogeny was inferred using RAxML-NG[69] with the GTR model, 20 starting trees (10 random and 10 parsimony), 100 bootstrap replicates, and a minimum branch length of 10−9. A Lineage 2 sample randomly selected from our dataset was selected as an outgroup for the Lineage 4 phylogeny. Likewise, a random Lineage 4 isolate was used as a root for the Lineage 2 phylogeny. The tree containing both lineage 4 and lineage 2 samples was rooted using a lineage 1 isolate.

In order to construct a time-calibrated phylogeny, we tested whether there was a detectable amount of evolutionary change between samples collected at different times[30,31]. This was done in lineage 4 and lineage 2 separately in order to avoid population structure confounding in the temporal signal[27]. Two different tests of the temporal signal were applied: the root-to-tip regression method and the date-randomization test[28]. For the former, BactDating[29] was used to perform a linear

regression between the collection dates and their root-to-tip genetic distance in the maximum likelihood tree. Additionally, we carried out a date-randomization test, where evolutionary rates estimated by BactDating[29] were compared between the observed data set and 100 data sets obtained by permutation of sampling dates[70].

BactDating[29] was used to time-calibrate the tree using the mixed model for $10^7$ iterations to achieve both convergence of the MCMC chains and an effective sample size of at least 100.

The phylogenetic global context of the Peruvian isolates was investigated by subsampling the phylogenies and repeating the analysis alongside publicly available isolates representative of the global diversity of *M. tuberculosis* (Supplementary Data File 2). To subsample the phylogenies, first a random sample was selected from each phylogenetic cluster with a branch length lower than 1 SNPs per genome. The phylogeny was then divided into clusters of samples 50 SNPs apart, and a maximum of 20 samples for lineage 2 and 5 samples for lineage 4 were randomly selected for each cluster, unless it consisted of only 1 sample, in which case it was ignored. The phylogeny with the Peruvian subsamples and the global representatives was inferred separately for lineage 2 and for lineage 4 as described above.

The subsequent phylogenetic analysis was performed using the R package ape[71]. Marginal reconstruction of the ancestral sequences was carried out by maximum likelihood as implemented in Phangorn[72], including gaps and ambiguity codes to reflect prior probabilities of character states[65].

**Time-to-event analysis.** Time-to-event analysis was performed on the tree using the R package Survival[73]. The time was measured for all pairs of nodes as the distance between the older and the younger node in the time-calibrated phylogeny. An observation was defined as censored if both nodes were drug sensitive. On the other hand, an event occurred if the older node was drug sensitive and the younger node was drug resistant. Only the first acquisition of resistance was considered. Observations taking place before 1940 were discarded. The Kaplan-Meier survival curve and the Cox proportional hazard ratio were calculated. The Kaplan-Meier curves for different groups were compared using the log-rank test, where the null hypothesis is that there is no difference in survival between the different groups. Differences in the hazard ratio were tested using the likelihood ratio test. The entire pipeline was repeated for 100 phylogenetic bootstrap replicates.

**Genome-wide association of predisposition to drug resistance.** Missing base calls at the tips were imputed by maximum likelihood using the re-rooting method[74] and the IUPAC ambiguous codes to reflect tip state prior probabilities. In short, for each tip the phylogeny was re-rooted at that tip and the marginal probabilities for the missing bases were calculated for that node using the R package phytools[75]. Association analysis was performed at the gene and SNP level using the variant sites alignment for the tips of the phylogenetic tree, as well as the reconstructed sequences of the internal nodes. The phenotype was defined as leading to resistance in the phylogenetic tree, and thus only drug susceptible nodes that immediately preceded the first resistant node of each branch were considered. For the gene level analysis, non-synonymous variants were aggregated, excluding lineage specific SNPs. Loci with a frequency of non-synonymous variants in the dataset lower than 1% were not considered in the analysis. At the SNP-level, variants with a frequency in the population (tips of the tree) lower than 1% were excluded. Furthermore, only those variants that were polymorphic at the node level were used. Genome-wide association was performed using the Cox proportional hazard model and the time between nodes. In order to correct for population structure, a genetic distance matrix was calculated using SNPs with a frequency in the population higher than 5%, and the eigenvectors were used as covariates in the Cox regression model. The genomic inflation factor ($\lambda$) was calculated as the ratio of the median of the empirically observed $\chi^2$ to the median of the expected $\chi^2$. The p-values were corrected for multiple testing using a Bonferroni correction. Functional annotation of the genomic variants was assessed using Mycobrowser[42]. Alignments were visually inspected for a random selection of samples using the Integrative genomics viewer (IGV)[76] and the R software package *Gviz*[77].

**Reporting summary.** Further information on research design is available in the Nature Research Reporting Summary linked to this article.

## Data availability

All raw sequencing data are available with accession numbers listed in the Methods section. Samples sequenced as part of this study have been submitted to the European Nucleotide Archive under accessions PRJEB39837 and PRJEB47846.

Publicly available datasets used in this study include PRJEB5280, PRJEB32234, PRJEB23245, and PRJEB2138.

All other publicly available datasets are listed in Supplementary Data File 2 and Supplementary Data File 3.

## Code availability

All custom code used in this article can be accessed at https://github.com/arturotorreso/mtb_pre-resistance.git.

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

## Acknowledgements

We would like to thank the participants of the study. LG was supported by the Wellcome Trust (201470/Z/16/Z), the National Institute of Allergy and Infectious Diseases of the National Institutes of Health under award number 1R01AI146338 and by the GOSH/ICH Biomedical Research Centre. OMK was supported by the Imperial Biomedical Research Centre (NIHR Imperial BRC, grant P45058). XD was supported by the NIHR Health Protection Research Unit in Genomics and Enabling Data. We thank the CRyPTIC project and the Tandem project for making whole-genome data available in the public domain. All authors acknowledge UCL Computer Science Technical Support Group (TSG) and the UCL Department of Computer Science High Performance Computing Cluster.

## Author contributions

A.T.O., L.G., X.D., O.M.K., F.B., J.R.V., R.G., C.B. and D.M. conceived and designed the study. J.C. performed and supervised tuberculosis cultures, DNA extraction, laboratory, and sequencing work. L.G., X.D., F.B. and O.M.K. jointly supervised the research. A.T.O., L.G., F.B. and X.D. performed and advised on computational analyses. A.T.O. and L.G. wrote the manuscript with input from all co-authors. All authors read and approved the final manuscript.

## Competing interests

The authors declare no competing interests.
