## [Peer Review File · Nature Communications]

Genomic signatures of pre-resistance in *Mycobacterium tuberculosis*REVIEWER COMMENTS

Reviewer #1 (Remarks to the Author):

This is a well conducted and very interesting piece, and the authors should be congratulated. They use a number of sound techniques to argue that lineage 2 strains acquire resistance faster than lineage 4 strains. They also identify 3 variants in lineage 4 that are associated with an increased risk of resistance acquisition.

I do however have a small number of questions based around two themes:

1. The claim sampling dates stretch over 17 years is true, but the spirit of the claim is somewhat undermined by the data presented in the appendix where the reader learns that almost 2/3 of samples were from a single year. What are the sampling years for L2 and L4 strains? In the introduction the authors outline the risk of 'insufficient temporal span' in a data set. Can they please reassure me that the extremely uneven sampling times in this data set overcome the described pitfalls? If not, this needs to be listed explicitly as a limitation in the discussion.

2. In line 261 the authors claim to have 'conclusively' shown that L2 acquired resistance faster than L4. I agree that a good case has been made but I would quibble with the claim that it is conclusive. There are potential confounders that have not been mentioned. For example, do we know if L2 and L4 affect different populations, such as different age groups? A lineage introduced more recently might be more strongly associated with younger patients. Different behaviours between patient groups could be a confounder. Do the authors have data on patient ages? Following on from this, is it known when L2 was introduced to Peru? The authors touched briefly on immigration patterns in the discussion, but is more known? It remains entirely possible that resistance in L2 was 'nurtured' in a different country, under different health care system conditions, and imported. The L2 tree appears quite narrow, suggesting relatively close genomic linkage between strains. Was there a founder effect for these samples? Some of this is touched upon in the discussion, but It would be helpful to say more.

Reviewer #2 (Remarks to the Author):

The manuscript Genomic signatures of Pre-Resistance in Mycobacterium tuberculosis was well written and will be of interest to both researchers in the field and to a wider audience. I have no major concerns about the results and these will be of some importance.

My only (minor) concerns are:

The sharing of sample metadata and associated accession numbers. Apart from project IDs that can be used to access the raw sequence data, unless I missed it, the authors have not provided sample metadata in the form of a supplementary file. This should be rectified before submission as this will be a dataset that will be useful to researchers around the world

The authors have not made any of their code available online. I would encourage the authors to avoid using a statement like "Variants that did not meet the quality criteria were filtered using a combination of BCFtools filter and custom scripts in Python v3.7.3" without making the code available for review. It would also be useful to see the R code used to perform the analyses in the study, particularly that which made use of the BactDating and Survival packages.

I'm unclear how the genome wide association analysis was performed. Did you use a package like pyseer or was this also done in R?

In the discussion the authors state "Here we demonstrate conclusively that lineage 2 acquired resistance to antibiotics more rapidly than lineage 4." Isn't this the other way round?

Figure 1: Would it be possible to show the 95% CI for each node as error bars as per BEAST?

Figure 2: In the legend the authors wrote "Arrows represent the approximated time of antibiotic distribution." Did they mean antibiotic introduction?

Reviewer #3 (Remarks to the Author):

The manuscript presents the analysis of a large collection of culture positive samples from Perú, some belonging to a population-based study in an area in Lima and the remaining belonging to sparse sampling in Perú over 17 years. The authors reconstruct the phylogenetic relationships between strains and then track the emergence of drug resistance associated mutation combining mutation mapping to the phylogenetic branches and different association analyses. The aim is to identify mutations predisposing for acquisition of additional resistance (pre-resistance). The authors identify several patterns including: 1) the role of mono-INH in amplifying drug resistance; 2) lineage 2 is associated with resistance and 3) identification of a region in *lppP* involved in pre-resistance.

While the approach used by the authors is novel (scanning a dated phylogenetic tree to identify emergence of resistance at the single nucleotide level) the results fall sort in terms of what is stated in the title and abstract. Below I will try to clarify my concerns and help the authors to improve the manuscript:

MAJOR

1.) While scientifically correct I don't find the novelty in two of the conclusions. INH precedes RIF and this has been known for a while. It is also one of the reasons why GenExpert focuses on RIF (the other is that mutations for RIF are highly concentrated in *rpoB* and have high explanatory power).

2) The authors focus a lot on the missing INH monoresistance cases when using Xpert (which only looks for RIF) (L277). However they don't explain that culture DST is still gold standard and carried out in many places (albeit with the known diagnostic delays problems). To be fair, authors should mention that INH monoresistance can be detected in most places using culture DST but given the diagnostic delays associated this maybe too late to avoid amplification of resistance

3) The other observation, lineage 2 is associated with resistance, is also known for a while although it is always difficult to distinguish between social and host factors of TB settings where lineage 2 predominates or pathogen factors. The present analysis shows in a controlled manner that under similar conditions lineage 2 is still associated with, what is a nice corroboration.

4) Sampling biases are of course always a danger in this kind of analysis. The authors recognized and showed how they match-and-mix different dataset to achieve enough temporal signal. However while the 2009 population-based analysis looks unbiased, probably is not the same for the rest of samples which are taken from convenience datasets. Can the authors break down in a table all datasets and specify the % susceptible, % rif monoresistance, % INH monoresistance, % MDR/XDR. A well balanced susceptible/resistance dataset is important to avoid over/under-representation of drug resistant associated branches.

5) The fact that the authors find a good correlation between introduction of a drug and their estimated timing of drug associated mutations is a nice proof of concept that their dating approach works. In the same lines, and if possible, can the authors comment on the timing for fitness compensatory mutations in *rpoC*? I think it will be novel to add some idea on when compensatory mutations started to circulate in the dataset and this is something that has not been shown before. In other words, are compensatory mutations associated branches arising shortly after the introduction of RIF or are they a more recent phenomenon and why?

6) Some questions worth exploring in your datasets. You say that rif mono-resistance is not frequent (L215). In many settings frequency is usually associated with relapse cases and also to rare *rpoB* mutations. When you say it is not frequent is that based on DST or on WGS predictions?. Rare mutations can cause false negative results (historically called "disputed mutation") in some systems and you may have missed some if not looking for those mutations?

7) In general, is there a good correlation between your DST and WGS-based DST?

8) I am still worried about a major assumption that may bias the analyses. The analysis assumes that the great majority of cases are not imported and thus not biasing the timing of emergence of drug resistance associated branches. But this is difficult to confirm as there is no data from the authors about this. Can

the authors specify how much information they have around imported cases in the dataset? I understand that particularly for the convenience samples it is difficult to tell. In that case maybe a phylogeny of the dataset with globally representative isolates may discern introductions (particularly those generating new drug resistance branches) from local emergence of resistance?

9) Not sure if survival analysis is adequate in this case, as it assumes that you start with a known number of individuals/populations and you can follow them through time. But here the starting population is unknowns and thus results will be highly biased by sampling. I am wondering how robust is to the use of subsets. For example if you downsample the 2009 dataset?

10) Then I will focus on the most interesting part of the work which is the identification of pre resistance mutations. What makes susceptible Mtb strains predisposed to acquire resistance mutations? The authors scan the phylogeny to identify such mutations in a novel approximation also combined with survival analysis. The authors identify lppP, esxO and an intergenic SNP. My first concern is that one of the two hits is a synonymous SNP in esxO. First, how do the authors explain a hit in a synonymous SNP? If this hit is spurious (as no effect is expected from a syn SNP) then it calls into question the approach to identify hits?. One explanation maybe that the hit generates a new transcriptional start site, is there any hint by looking at the sequence data that a new regulatory region has been created? Is it homoplastic (what will reinforce the notion that is associated with selection)? Alternatively, esx family gene members align poorly when using short reads because repeated sequences along the genome. Can the authors look at individual alignments and confirm there is no such error here? Otherwise how do the authors explain a synonymous hit?

11) About lppP- it is only present in 1.7%. By itself has a low explanatory power and not sure if enough to justify the title of the manuscript (which is basically based on this hit).

12) In addition. This hit and others cannot be validated using the Russian dataset which is understandable given the sample size and that the genetic make up of the Russian dataset is really different. Is there any evidence (even if not statistically significant) for the presence of any of the hits in the Russian dataset?

13) However an extraordinary claim needs extraordinary evidence. I suggest the authors look for alternative validations of the hit. I can think in several ways but one relatively easy (if you have access to the isolates) is to measure and show that isolates carrying that particular mutation have a higher MIC for INH than phylogenetically related isolates without the mutation. Alternatively authors may look for publications in which mutant libraries (with transposon) are tested against first line drugs. Authors can also identify alternative datasets where this can be tested as there are many around nowadays or at the very least authors can look in the global database and see the prevalence of the mutation across continents. The ideal validation will be a KO mutant or an allelic mutant but I understand this is extraordinarily difficult and time-consuming in the case of Mtb.

14) In general, and maybe reflecting my ignorance, looks like p-values in the analysis of pre-resistance mutations are extremely low. This suggests a clear link and in this case also a strong effect. This is good but given that many groups have been working on this and had problems to identify these regions I wonder where the difference resides to obtain such large effects?

15) Regarding the Russian dataset I don't see the full value of incorporating the analysis. It confirms some of the findings but looks like it has more problems than advantages for your purposes. It cannot be used to validate as discussed above. But also it is not dated and thus it is difficult to compare to the main dataset like in Fig4 where one panel is based on years and the other on branch lengths. It is not possible to compare the trajectories in such a case.

16) Even if comparable the Kaplan Meier curve is quite different in both cases and I am struggling to understand why? One explanation is different biases in the dataset (being enriched in MDR/XDR and very few sensitive and monoresistant?). In general Figure 4 should be in the same scale so it can be properly compared and interpret

Minor

1) Can you specify the substitution rate obtained for each lineage and see if it compares well with published rates?

- 2)L127. The authors should clarify that 1% frequency refers to the SNP in the analysed dataset and not in the bulk sequencing of individual cultures
- 3)L428. Additional details on the ancestral reconstruction algorithm would be welcome, i.e. reversion penalties

Response to reviewers' comments:

We would like to thank the reviewers for their time and insightful and constructive comments. We believe the paper has been improved as a result, which is reflected in the fact that many of their suggestions have now been included as main figures, tables or entire sections.

REVIEWER COMMENTS

Reviewer #1 (Remarks to the Author):

This is a well conducted and very interesting piece, and the authors should be congratulated. They use a number of sound techniques to argue that lineage 2 strains acquire resistance faster than lineage 4 strains. They also identify 3 variants in lineage 4 that are associated with an increased risk of resistance acquisition.

I do however have a small number of questions based around two themes:

R1.1. The claim sampling dates stretch over 17 years is true, but the spirit of the claim is somewhat undermined by the data presented in the appendix where the reader learns that almost 2/3 of samples were from a single year. What are the sampling years for L2 and L4 strains? In the introduction the authors outline the risk of 'insufficient temporal span' in a data set. Can they please reassure me that the extremely uneven sampling times in this data set overcome the described pitfalls? If not, this needs to be listed explicitly as a limitation in the discussion.

The reviewer raises an important point regarding the possible biases when calibrating a phylogeny using the time of collection of the isolates at the tips of the tree. As the reviewer points out, most of our samples were collected as part of a population level study in 2009-2010. Even though the root-to-tip regression for assessing temporal signal can be biased by a very uneven sampling time, by performing the date-randomization test we clearly show that we can confidently infer evolutionary parameters using our phylogeny, an evolutionary model and the study's sampling window when compared to a random permutation of sampling dates. We discuss this point further in the Results section, "Phylogenetic analysis and drug resistance emergence" subsection, line 151: "Before time calibration of the phylogeny, the presence of enough evolutionary change to reliably infer the model parameters was tested. First, a linear regression of the number of substitutions from the root and the sampling times was fitted to confirm a positive association between time and evolutionary change. As an uneven sampling may bias the root-to-tip regression [28], a date-randomization test was additionally performed using the full Bayesian model implemented in BactDating [29] on the original dataset and in 100 randomizations where the sampling times were permuted, representing the expectations of the model parameters in the absence of temporal signal. The substitution rate estimated for the original dataset and for the 100 randomizations was compared to verify a lack of overlap between the 95% credible intervals." The confirmation of a temporal signal both with linear regression and with BactDating permutation as described make us confident that we have overcome any bias due to uneven temporal sampling.

As suggested, to show lineage-specific sampling times we have now added the distribution of sampling times as histograms separately for lineage 2 (L2) and Lineage 4 (L4) as Supplementary Figure 2, referenced on line 115: “Both lineage 2 and lineage 4 showed a similar distribution of sampling dates (Supplementary Fig. 2)”

R1.2. In line 261 the authors claim to have 'conclusively' shown that L2 acquired resistance faster than L4. I agree that a good case has been made but I would quibble with the claim that it is conclusive. There are potential confounders that have not been mentioned. For example, do we know if L2 and L4 affect different populations, such as different age groups? A lineage introduced more recently might be more strongly associated with younger patients. Different behaviours between patient groups could be a confounder. Do the authors have data on patient ages?

To account for some of the potential confounders described by the reviewer, we have additionally explored the age distribution of the patients for L2 and L4 in a linear model. The mean age for patients with L4 was 33.3 (22-41 IQR), while for L2 the mean age was 30.4 (21-36 IQR). To explore the age differences further, we fitted a quasi-Poisson model, showing an incident rate on age higher for lineage 4 when compared to lineage 2, though with a very low estimate (lineage 4 estimate = 0.09, $p = 0.001$). Additionally, we explored whether age was associated to a higher risk of drug resistance acquisition by fitting a logistic regression model, and showed that age was not significantly associated with a higher risk of drug resistance acquisition in a logistic model (OR 0.999, 95%CI 0.994-1.003, $p=0.69$). We are grateful to the reviewer for this pertinent comment and feel that by addressing it with this further analysis we strengthened the paper. Indeed, given these new results, it is not probable that age is an important confounder in drug resistance acquisition for our dataset. This is reported in Results, subsection “Between lineage differences in drug resistance acquisition”, paragraph 4, as well as supplementary figure 9.

In spite of the potential confounders included in the analysis, we agree with the reviewer that there can be other cofactors influencing the patterns we observe. Therefore, as suggested, we removed the word “conclusively” from the discussion.

R1.3 Following on from this, is it known when L2 was introduced to Peru? The authors touched briefly on immigration patterns in the discussion, but is more known? It remains entirely possible that resistance in L2 was 'nurtured' in a different country, under different health care system conditions, and imported. The L2 tree appears quite narrow, suggesting relatively close genomic linkage between strains. Was there a founder effect for these samples? Some of this is touched upon in the discussion, but It would be helpful to say more.

We agree with the reviewer that this is an important point, and therefore we added a phylogenetic tree showing the global context of our Peruvian dataset alongside publicly available samples from different countries. We subsetted the dataset as described in the Methods section, line L586. The results are shown in figure 4 and the subsection “Phylogeographic history of Mycobacterium tuberculosis in Peru”. We time calibrated the phylogeny and inferred the time of introduction, which for the biggest clades predated the

advent of antibiotics. Given this analysis, we concluded that importation of drug resistant strains from different countries and health systems has not been wide enough to explain the differential patterns in drug resistance acquisition between lineage 2 and lineage 4. Apart from this, we acknowledge we may be missing important key importation events, and therefore we caution that this could still affect our results in the Discussion.

Reviewer #2 (Remarks to the Author):

The manuscript Genomic signatures of Pre-Resistance in Mycobacterium tuberculosis was well written and will be of interest to both researchers in the field and to a wider audience. I have no major concerns about the results and these will be of some importance.

My only (minor) concerns are:

R2.1 The sharing of sample metadata and associated accession numbers. Apart from project IDs that can be used to access the raw sequence data, unless I missed it, the authors have not provided sample metadata in the form of a supplementary file. This should be rectified before submission as this will be a dataset that will be useful to researchers around the world

Sample metadata is now provided as part of Supplementary Data File 1. Any other metadata will be available upon request.

R2.2 The authors have not made any of their code available online. I would encourage the authors to avoid using a statement like “Variants that did not meet the quality criteria were filtered using a combination of BCFtools filter and custom scripts in Python v3.7.3” without making the code available for review. It would also be useful to see the R code used to perform the analyses in the study, particularly that which made use of the BactDating and Survival packages.

We agree with the reviewer on the importance of making code available, which was missing in our previous manuscript. All custom code is publicly available on GitHub (https://github.com/arturotorreso/mtb_pre-resistance.git), and the link is properly referenced in both Methods and Code Availability sections.

R2.3 I'm unclear how the genome wide association analysis was performed. Did you use a package like pyseer or was this also done in R?

The Cox regression GWAS was done in R, using the survival package. The code is available on GitHub https://github.com/arturotorreso/mtb_pre-resistance.git.

R2.4 In the discussion the authors state “Here we demonstrate conclusively that lineage 2 acquired resistance to antibiotics more rapidly than lineage 4.” Isn't this the other way round?

The statement highlighted by the reviewer is correct: the results show that lineage 2 acquires resistance faster than lineage 4, which is shown in the Kaplan-Meier curves where the probability of remaining susceptible decreases earlier and faster for lineage 2.

R2.5 Figure 1: Would it be possible to show the 95% CI for each node as error bars as per BEAST?

To address this point we have included confidence intervals of the most relevant nodes (e.g, root, drug resistance nodes) in the main text and tables. Given the size of the phylogenetic tree, most nodes are very close to each other and thus adding confidence intervals onto nodes in the tree would make the graphic noisy and difficult to interpret.

R2.6 Figure 2: In the legend the authors wrote "Arrows represent the approximated time of antibiotic distribution." Did they mean antibiotic introduction?

Yes, although we acknowledge that the time of introduction of various drugs to different countries may vary, as well as the time for widespread use. Finding such historical data can be challenging, but the confidence interval around resistance emergence estimates should encompass the wide range of possible drug introduction dates. We have changed the legend of Figure 2 to use the more appropriate term "Introduction", as suggested by the reviewer.

Reviewer #3 (Remarks to the Author):

The manuscript presents the analysis of a large collection of culture positive samples from Perú, some belonging to a population-based study in an area in Lima and the remaining belonging to sparse sampling in Perú over 17 years. The authors reconstruct the phylogenetic relationships between strains and then track the emergence of drug resistance associated mutation combining mutation mapping to the phylogenetic branches and different association analyses. The aim is to identify mutations predisposing for acquisition of additional resistance (pre-resistance). The authors identify several patterns including: 1) the role of mono-INH in amplifying drug resistance; 2) lineage 2 is associated with resistance and 3) identification of a region in *lppP* involved in pre-resistance.

While the approach used by the authors is novel (scanning a dated phylogenetic tree to identify emergence of resistance at the single nucleotide level) the results fall short in terms of what is stated in the title and abstract. Below I will try to clarify my concerns and help the authors to improve the manuscript:

MAJOR

R3.1 While scientifically correct I don't find the novelty in two of the conclusions. INH precedes RIF and this has been known for a while. It is also one of the reasons why GenExpert focuses on RIF (the other is that mutations for RIF are highly concentrated in *rpoB* and have high explanatory power).

As the reviewer points out, several phylogenetic studies acknowledged in our work have shown that the emergence of INH resistance tends to precede that of RIF. But in our view those studies do not show that there is a higher risk of additional RIF resistance when isolates carry mutations that confer INH resistance, which our work shows by directly comparing the risk of emergence of RIF between INH-Resistance isolates and drug-susceptible ones. INH may historically have preceded RIF emergence because of other factors, including the earlier introduction of INH as

treatment for tuberculosis. Our findings highlight that the likely consequence of the failing to detect INH monoresistance with the GenExpert is the rapid evolution of MDRTB.

R3.2 The authors focus a lot on the missing INH monoresistance cases when using Xpert (which only looks for RIF) (L277). However they don't explain that culture DST is still gold standard and carried out in many places (albeit with the known diagnostic delays problems). To be fair, authors should mention that INH monoresistance can be detected in most places using culture DST but given the diagnostic delays associated this maybe too late to avoid amplification of resistance

We have now changed the discussion to reflect this point, specifically line L401: "Globally, rapid drug resistance surveillance is focused primarily on rifampicin, with the widely implemented GeneXpert MTB/RIF PCR based assay unable to detect isoniazid mono-resistance. Although drug susceptibility testing (DST) is the current gold standard for identification of drug resistance isolates and can detect isoniazid mono-resistance, known diagnostic delays associated with it may limit its use in reducing mono-resistance amplification [40,41]."

R3.3 The other observation, lineage 2 is associated with resistance, is also known for a while although it is always difficult to distinguish between social and host factors of TB settings where lineage 2 predominates or pathogen factors. The present analysis shows in a controlled manner that under similar conditions lineage 2 is still associated with, what is a nice corroboration.

We agree with the reviewer that some studies, both laboratory and epidemiological ones, have shown a differential rate of drug resistance acquisition between lineages, but many analyses have also contradicted this idea (eg., Glynn et al. 2002, Werngren et al. 2003). In our view, it remained unclear whether there was a lineage effect on drug resistance acquisition at the population level and over a large time span, and therefore we believe our analysis represents more than a corroboration of earlier results.

1. Glynn, J. R., Whiteley, J., Bifani, P. J., Kremer, K. & van Soolingen, D. Worldwide Occurrence of Beijing/W Strains of Mycobacterium tuberculosis : A Systematic Review. *Emerging Infectious Diseases* 8,843–849 (2002).
2. Werngren, J. & Hoffner, S. E. Drug-susceptible Mycobacterium tuberculosis Beijing genotype does not develop mutation-conferred resistance to rifampin at an elevated rate. *Journal of Clinical Microbiology* 41,1520–1524 (2003). 13

R3.4 Sampling biases are of course always a danger in this kind of analysis. The authors recognized and showed how they match-and-mix different dataset to achieve enough temporal signal. However while the 2009 population-based analysis looks unbiased, probably is not the same for the rest of samples which are taken from convenience datasets. Can the authors break down in a table all datasets and specify the % susceptible, % rif monoresistance, % INH monoresistance, % MDR/XDR. A well balanced susceptible/resistance dataset is important to avoid over/under-representation of drug resistant associated branches.

As the reviewer points out, a dataset enriched for resistance may bias the results. We have now added a supplementary table (supplementary table 1) with the information required by the reviewer regarding the rates of drug resistance isolates in each study. We have also replicated our analysis in a tree with only the 2009 population level dataset to confirm our results

(supplementary figure 9). Our results show that, even though part of the dataset is enriched for drug resistance samples (in particular multi-drug resistance ones), the inclusion of a sufficiently high number of drug susceptible isolates reduces the potential biases.

R3.5 The fact that the authors find a good correlation between introduction of a drug and their estimated timing of drug associated mutations is a nice proof of concept that their dating approach works. In the same lines, and if possible, can the authors comment on the timing for fitness compensatory mutations in *rpoC*? I think it will be novel to add some idea on when compensatory mutations started to circulate in the dataset and this is something that has not been shown before. In other words, are compensatory mutations associated branches arising shortly after the introduction of RIF or are they a more recent phenomenon and why?

The reviewer raises an interesting point, and we agree that it would be relevant to understand the temporal dynamics of well known compensatory mutations, which could in turn help find novel ones. We have thus added a new subsection to Results: “The emergence of compensatory mutations”, and a main figure (Figure 3). As the reviewer suggested, and since it's the best (if only) well known compensatory mechanism, we focused on non-synonymous mutations in the *rpoC* gene. We analyzed the emergence of non-synonymous *rpoC* mutations along the phylogeny, and found that the vast majority of mutations appear shortly after rifampicin resistance has been acquired, and keep accumulating through time.

R3.6 Some questions worth exploring in your datasets. You say that rif mono-resistance is not frequent (L215). In many settings frequency is usually associated with relapse cases and also to rare *rpoB* mutations. When you say it is not frequent is that based on DST or on WGS predictions?. Rare mutations can cause false negative results (historically called “disputed mutation”) in some systems and you may have missed some if not looking for those mutations?

Acknowledging that rifampicin resistance emergence can be caused by rare mutations, we compared the incidence of rifampicin resistance by MODS or by the proportion method in agar. The proportion of RIF mono resistance by DST was 2.77% (87/3134) and the proportion of RIF mono resistance by genotypic determination was 2.1% (65/3134). The differences between the two proportions are mostly due to the presence of isoniazid resistance conferring mutations when using molecular genotyping as has previously been described. Moreover, DST and molecular typing were congruent in 96% of the samples for rifampicin resistance. Given the strong correlation between phenotypic and genotyping resistance testing for Rif we believe that the impact of rare Rif mutations on our findings is not significant. This is now reflected in the text, Results section, subsection “Phylogenetic analysis and drug resistance emergence”, paragraph 2: “In addition to molecular typing of drug resistance, all isolates included in the analysis had drug susceptibility testing performed either by MODS or by the proportional method in agar. DST and molecular typing showed consistent results for 96% of the samples for rifampicin resistance and 92% for isoniazid”

R3.7 In general, is there a good correlation between your DST and WGS-based DST?

DST and molecular typing showed consistent results for 96% of the samples for rifampicin resistance and 92% for isoniazid. Most differences are caused by isoniazid resistance conferring mutations detected by molecular typing but showing a susceptible phenotype by DST.

R3.8 I am still worried about a major assumption that may bias the analyses. The analysis assumes that the great majority of cases are not imported and thus not biasing the timing of emergence of drug resistance associated branches. But this is difficult to confirm as there is no data from the authors about this. Can the authors specify how much information they have around imported cases in the dataset? I understand that particularly for the convenience samples it is difficult to tell. In that case maybe a phylogeny of the dataset with globally representative isolates may discern introductions (particularly those generating new drug resistance branches) from local emergence of resistance?

We strongly agree with the reviewer that the publication will greatly improve if we address the assumption that all drug resistance is acquired in Peru. Therefore, we have now included a phylogeny containing representatives of our lineages alongside a publicly available global dataset of *Mycobacterium tuberculosis* (Supplementary Data File 2). The results are shown in Figure 4 and subsection "Phylogeographic history of *Mycobacterium tuberculosis* in Peru". In short, the main clades of our dataset are the result of introductions pre-dating the advent of antibiotics, and thus the influence of imported resistance is likely to be limited. Nevertheless, as stated in the Discussion, a possible bias cannot be completely ruled out.

R3.9 Not sure if survival analysis is adequate in this case, as it assumes that you start with a known number of individuals/populations and you can follow them through time. But here the starting population is unknown and thus results will be highly biased by sampling. I am wondering how robust is to the use of subsets. For example if you downsample the 2009 dataset?

We agree with the reviewer that the method could potentially be biased by very uneven sampling of isolates, especially in drug resistance clusters. We repeated the analysis only on the population level 2009 dataset to address R3.4 point, and found similar results than when using the entire dataset.

We use the last sensitive speciation event as the starting population for each occurrence of drug resistance. The proportional hazard assumption implies that hazard ratios are constant over time, and therefore as long as this assumption is not violated the method would still be valid even if the starting time is not completely known.

R3.10 Then I will focus on the most interesting part of the work which is the identification of pre resistance mutations. What makes susceptible *Mtb* strains predisposed to acquire resistance mutations? The authors scan the phylogeny to identify such mutations in a novel approximation also combined with survival analysis. The authors identify *lppP*, *esxO* and an intergenic SNP. My first concern is that one of the two hits is a synonymous SNP in *esxO*. First, how do the authors explain a hit in a synonymous SNP? If this hit is spurious (as no effect is expected from a syn SNP) then it calls into question the approach to identify hits?. One explanation maybe that the hit generates a new transcriptional start site, is there any hint by looking at the sequence data that a new regulatory region has been

created? Is it homoplastic (what will reinforce the notion that is associated with selection)? Alternatively, esX family gene members align poorly when using short reads because repeated sequences along the genome. Can the authors look at individual alignments and confirm there is no such error here? Otherwise how do the authors explain a synonymous hit?

As pointed out in the Discussion, synonymous homoplastic polymorphisms in esx genes have been identified in the past, but we agree their relevance is still unknown, and further studies (including gene expression assays) should be performed.

Regarding the quality of the alignments, we have now added Supplementary figure 12 showing the alignment of the short-reads around the GWAS hits. The visual inspection suggests the alignments are of good quality.

R3.11 About lppP- it is only present in 1.7%. By itself has a low explanatory power and not sure if enough to justify the title of the manuscript (which is basically based on this hit).

To increase the number of putative regions associated with drug resistance in inferred susceptible nodes, we have complemented the SNP based GWAS with a gene-based analysis, aggregating non-synonymous mutations. This method has been proved useful in finding drug resistance associated mutations in *Mycobacterium tuberculosis* (see Coll et al 2018 and Farhat et al 2019). We have also added a step of sequence base imputation to account for SNPs that may have been overlooked in the previous analysis. Overall, we think that these associations can be used as a “proof of concept”, but we agree with the reviewer that further analyses are needed, which is now also noted in the Discussion section.

Regarding lppP, we confirmed its presence in a global dataset (Supplementary Data File 3) where its frequency was as high as 9%, indicating its global prevalence and suggesting a role in drug resistance acquisition. Such a big difference in frequency with our dataset may either be due to the mutation not being very prevalent in Peru, or due to the global dataset being more enriched with drug resistant isolates. We replicated the analysis on the global dataset, where the lppP deletion was associated with a higher risk of drug resistance in inferred susceptible nodes.

1. Coll, F., Phelan, J., Hill-Cawthorne, G.A. et al. Genome-wide analysis of multi- and extensively drug-resistant *Mycobacterium tuberculosis*. *Nat Genet* 50, 307–316 (2018).
2. Farhat, M.R., Freschi, L., Calderon, R. et al. GWAS for quantitative resistance phenotypes in *Mycobacterium tuberculosis* reveals resistance genes and regulatory regions. *Nat Commun* 10, 2128 (2019).

R3.12 In addition. This hit and others cannot be validated using the Russian dataset which is understandable given the sample size and that the genetic make up of the Russian dataset is really different. Is there any evidence (even if not statistically significant) for the presence of any of the hits in the Russian dataset?

We replicated the analysis in a global dataset of 1573 publicly available isolates (see R3.11), which was enriched with drug resistance samples. The frequencies of all the hits, as well as the results of the analysis, are discussed in the Results section, subsection “Genomic signatures of drug resistance acquisition”, paragraph 2. In short, The *lppP* deletion had a frequency of 9% and had a hazard ratio 3.6 times greater than those without the deletion (HR 3.6, 95% CI 1.9-6.9, p-value = 8.7e-5). The *esxO* mutation had a frequency of 10% had a risk of acquiring drug resistance 3.1 times higher than those with the reference genotype (HR 3.1, 95% CI 1.3-7.3, p-value = 0.009).

R3.13 However an extraordinary claim needs extraordinary evidence. I suggest the authors look for alternative validations of the hit. I can think in several ways but one relatively easy (if you have access to the isolates) is to measure and show that isolates carrying that particular mutation have a higher MIC for INH than phylogenetically related isolates without the mutation. Alternatively authors may look for publications in which mutant libraries (with transposon) are tested against first line drugs. Authors can also identify alternative datasets where this can be tested as there are many around nowadays or at the very least authors can look in the global database and see the prevalence of the mutation across continents. The ideal validation will be a KO mutant or an allelic mutant but I understand this is extraordinarily difficult and time-consuming in the case of *Mtb*.

We agree with the reviewer that further work and evidence it's needed to validate these hits as well as to add future genotypes.

Given that our aim is to identify polymorphisms in drug sensitive backgrounds that increase the risk of acquiring drug resistance; we hypothesize that comparing the MIC's of closely related mutant/wild type strains in-vitro is unlikely to demonstrate a difference in MIC's. Rather our differences are more likely to manifest in vivo when exposed to selection pressure from drugs and host immunity combined. Despite the limitations of in vitro work, we are working on laboratory confirmation of these mutations using fluctuation assays, as well as GWAS analysis in global and more diverse datasets.

As the reviewer suggests, we have also checked the prevalence of the main GWAS hits in the global dataset used to time the importation events of TB in Peru. This is described in the Results section, subsection “Genomic signatures of drug resistance acquisition”, paragraph 2 (see R3.11 and R3.12). In short, the three polymorphisms described in our work are present in the global dataset and in higher frequency. This could be explained by the higher proportion of drug resistance isolates in the publicly available data. Both *lppP* and *esxO* were associated with a higher risk of drug resistance in the global data set. These points have been reflected in the Results and Discussion.

R3.14 In general, and maybe reflecting my ignorance, looks like p-values in the analysis of pre-resistance mutations are extremely low. This suggests a clear link and in this case also a strong effect. This is good but given that many groups have been working on this and had problems to identify these regions I wonder where the difference resides to obtain such large effects?

We are not aware of any group working on this concept, especially in a phylogenetic context. We believe the differences may be caused by the presence of more susceptible samples in our dataset. Given that we are looking for mutations in genotypes inferred to be susceptible, a large number of susceptible isolates is required for the inference. Many GWAS collections are very enriched with drug- and multidrug-resistant samples, and therefore they may lack sufficient resolution to infer the susceptible ancestral genotypes correctly. For instance, many of the mutations described in our work may occur alongside drug resistance conferring mutations, and therefore they will be masked by them.

R3.15 Regarding the Russian dataset I don't see the full value of incorporating the analysis. It confirms some of the findings but looks like it has more problems than advantages for your purposes. It cannot be used to validate as discussed above. But also it is not dated and thus it is difficult to compare to the main dataset like in Fig4 where one panel is based on years and the other on branch lengths. It is not possible to compare the trajectories in such a case. Even if comparable the Kaplan Meier curve is quite different in both cases and I am struggling to understand why? One explanation is different biases in the dataset (being enriched in MDR/XDR and very few sensitive and monoresistant?). In general Figure 4 should be in the same scale so it can be properly compared and interpret.

The Samara dataset represents an interesting comparison, as it is the other largest population level project in a setting where lineage 2 is the major lineage. We acknowledge that the different proportion of drug resistance between datasets makes them harder to compare, but even so, the Samara dataset and the Peruvian one both show that lineage 2 acquires resistance faster than lineage 4, either when measured in years or genetic distance.

Minor

R3.m1 Can you specify the substitution rate obtained for each lineage and see if it compares well with published rates?

The posterior density distribution for the substitution rate for each lineage is shown as violin plots in Figure 1b. As pointed out in Results, subsection "Phylogenetic analysis and drug resistance emergence", line 151-152 of the original manuscript, these estimates are consistent with previous published ones.

R3.m2 L127. The authors should clarify that 1% frequency refers to the SNP in the analysed dataset and not in the bulk sequencing of individual cultures

The sentence at line L127 has now been changed to: "Most SNPs were not widely distributed across the population, and only 8088 variants had a frequency in the dataset higher than 1%"

R3.m3 L428. Additional details on the ancestral reconstruction algorithm would be welcome, i.e. reversion penalties

Marginal ancestral states for the sequences were estimated by maximum likelihood by optimizing the transition rate matrix and the rates and finding the state at each node that maximizes the likelihood of the data. This has now been clarified in Methods.

REVIEWERS' COMMENTS

Reviewer #1 (Remarks to the Author):

The authors have given extensive, well thought through and well evidenced replies to my comments. I am satisfied with the changes that have been made. I have no further comments.

Reviewer #3 (Remarks to the Author):

The authors have addressed the concerns raised by the reviewers. Particularly important is the validation in another cohort. Some very minor comments

1) Their evolutionary approach to detect "pre-resistance" is sound and complements efforts to identify candidate loci and mutations associated to higher likelihood to develop additional resistance, for example through mechanisms like drug tolerance mediated by prpR which the authors may want to cite (<https://www.nature.com/articles/s41564-018-0218-3>).

2) I also thank the authors their new analysis about compensatory mutations. A minor comments is whether the same early identification of rpoC mutations remain true when you disgregate the data between S450L and non-S450L rpoB mutation. The reason is that there has been observed a strong association between S450L and compensatory mutations although S450L has by far the lowest fitness cost of all rpoB mutations. The reasons are not clear (one hypothesis is that only those mutations with small fitness costs can be compensated while the others are beyond compensation). A different emergence dynamics in the two subgroups will reinforce the view of S450L being preferentially compensated. It is just a suggestions as I know non-S450L compensated strains are not that common.

Response to reviewers' comments:

Reviewer #1 (Remarks to the Author):

The authors have given extensive, well thought through and well evidenced replies to my comments. I am satisfied with the changes that have been made. I have no further comments.

Reviewer #3 (Remarks to the Author):

The authors have addressed the concerns raised by the reviewers. Particularly important is the validation in another cohort. Some very minor comments

1) Their evolutionary approach to detect "pre-resistance" is sound and complements efforts to identify candidate loci and mutations associated to higher likelihood to develop additional resistance, for example through mechanisms like drug tolerance mediated by prpR which the authors may want to cite (<https://www.nature.com/articles/s41564-018-0218-3>).

We have now included this relevant reference in the 5th paragraph of the Discussion. We thank the reviewer for pointing out conditional drug tolerance as another mechanism of increasing the risk of drug resistance acquisition.

2) I also thank the authors their new analysis about compensatory mutations. A minor comments is whether the same early identification of rpoC mutations remain true when you disgregate the data between S450L and non-S450L rpoB mutation. The reason is that there has been observed a strong association between S450L and compensatory mutations although S450L has by far the lowest fitness cost of all rpoB mutations. The reasons are not clear (one hypothesis is that only those mutations with small fitness costs can be compensated while the others are beyond compensation). A different emergence dynamics in the two subgroups will reinforce the view of S450L being preferentially compensated. It is just a suggestions as I know non-S450L compensated strains are not that common.

We agree with the reviewer that it would be interesting to analyze the dynamics of emergence of compensatory mutations in S450L strains vs non-S450L ones. As the reviewer points out, we confirmed that isolates with S450L rpoB had a higher probability of acquiring rpoC mutations when compared to non-S450L. This has been added to the subsection "The emergence of compensatory mutations", first paragraph: "Overall, 62% of rifampicin resistant isolates carried Ser450Leu rpoB mutations (525/842). Rifampicin resistant isolates harboring Ser450Leu rpoB mutations had a higher probability of carrying non-synonymous mutations in the rpoC gene (52%, 272/525) than isolates with other rpoB mutations (6%, 19/317) in a logistic regression model (OR=16.86, 95% CI 10.55-28.50, p<0.001)". On the other hand, both Ser450Leu and non-Ser450Leu showed similar temporal dynamics, as it is now shown in the new Supplementary Figure 6, although with high uncertainty given the low number of non-Ser450Leu rifampicin resistant isolates.